# Logical Consistency of Large Language Models in Fact-Checking

**Bishwamittra Ghosh[1, *], Sarah Hasan[2, *], Naheed Anjum Arafat[3], Arijit Khan[2]**

[1]Max Planck Institute for Software Systems, Germany

[2]Aalborg University, Denmark

[3]Independent Researcher, USA

[1]bghosh@mpi-sws.org, [2]{sarahh, arijitk}@cs.aau.dk, [3]naheed_anjum@u.nus.edu

## Abstract

In recent years, large language models (LLMs) have demonstrated significant success in performing varied natural language tasks such as language translation, question-answering, summarizing, fact-checking, etc. Despite LLMs' impressive ability to generate human-like texts, LLMs are infamous for their *inconsistent* responses – a meaning-preserving change in the input query results in an inconsistent response and attributes to vulnerabilities of LLMs such as hallucination. Consequently, existing research focuses on simple paraphrasing-based consistency assessment of LLMs, and ignores complex queries that necessitate an even better understanding of logical reasoning by an LLM. *Our work therefore addresses the logical inconsistency of LLMs under complex logical queries with primitive logical operators, e.g., negation, conjunction, and disjunction.* As a test bed, we consider retrieval-augmented LLMs on a fact-checking task involving propositional logic queries from knowledge graphs (KGs). Our contributions are threefold. **Benchmark:** We introduce three logical fact-checking datasets over KGs for community development towards logically consistent LLMs. **Assessment:** We propose consistency measures of LLMs on propositional logic queries and demonstrate that existing LLMs lack logical consistency, especially on complex queries. **Improvement:** We employ supervised fine-tuning to improve the logical consistency of LLMs on the complex fact-checking task with KG contexts. We have made our source code and benchmarks available[1].

## 1 Introduction

Large language models (LLMs), e.g., ChatGPT (OpenAI, 2022), PaLM (Chowdhery et al., 2023), LLaMA (Touvron et al., 2023) permeate natural language processing with their remarkable capacity to comprehend and generate human-like texts (Min et al., 2023; Kamalloo et al., 2023). As such, there is a host of applications of LLMs in high-stake domains such as healthcare (Singhal et al., 2023; Wang et al., 2023b; Dash et al., 2023), finance (Wu et al., 2023; Yang et al., 2023a), law (Xiao et al., 2021), and education (Kasneci et al., 2023). For instance, by implicitly storing pre-trained/fine-tuned knowledge, LLMs are increasingly being applied in fact-checking and question-answering tasks (Saeed et al., 2023; Quelle & Bovet, 2024).

**Consistency of LLM Response.** Despite the success story, a notable challenge in LLMs is the *inconsistency* of generated responses. Consistency of an LLM – the invariance of its output under meaning-preserving alternations in its input – is a highly desirable property (Elazar et al., 2021; Liu et al., 2023c). Existing works mostly focus on semantically similar sentences (paraphrases) for consistency assessment. For example, Kuhn et al. (2023) measures the consistency of LLMs by computing semantic uncertainty of output responses under paraphrased inputs. Elazar et al. (2021) consider quasi-paraphrases in a cloze-style input format and assess masked language models (Devlin et al., 2018) for consistency (additional related works are discussed in the Appendix B).

---

*Jointly first authors.

[1]https://github.com/bishwamittra/llm_logical_consistency

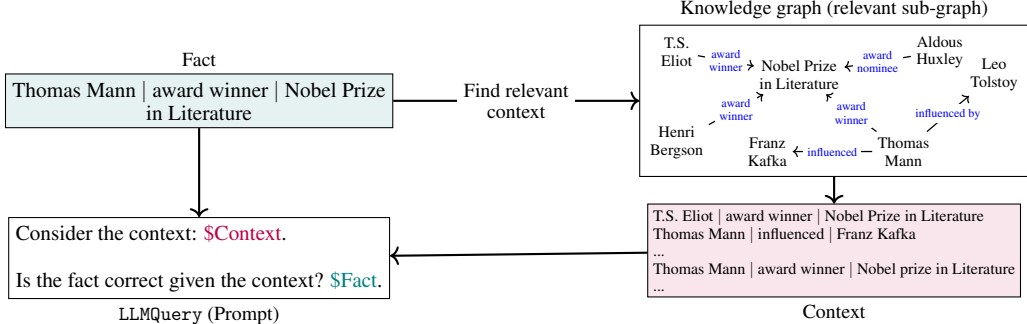

Figure 1: Our LLM-based fact-checking framework on a simple fact with context from a knowledge graph. A representative LLMQuery is in Figure 3, and the extension to complex facts is in Figure 4.

Existing works focus on the consistency of an LLM on simple queries and do not assess consistency on complex queries requiring improved logical reasoning by LLMs. Therefore, an unexplored research question is to systematically assess and improve the logical consistency of LLMs in complex logic queries with primitive logical operators: negation ($\neg$), conjunction ($\wedge$), and disjunction ($\vee$). Informally, an LLM is logically consistent with a query under the negation operator if the LLM response to the base query and the negated query are semantically opposite. Subsequently, an LLM is consistent on a conjunction query with (at least) two sub-queries if the individual responses to each sub-queries are logically consistent with the conjunction query. Building on logical semantics, we naturally extend the consistency assessment of LLMs to propositional logic queries and rules such as commutative, distributive, associative, syllogism, and to first-order logic queries (Boole, 1847). Our hypothesis is that logically consistent LLMs are essential in trustworthy systems (Sun et al., 2024b): When an LLM is consistent, it is easier to verify whether the LLM is right or wrong, without a sophisticated benchmark – being logically consistent provides the end users additional confidence in applying LLMs in their own workload.

**Logical Consistency in Fact-Checking via Retrieval-Augmented Generation.** In this paper, we assess and improve the logical consistency of LLMs in propositional logic queries and logic rules, particularly in the context of fact-checking. Automated fact-checking is the task of deploying an intelligent system such as an LLM to verify a queried fact with respect to a baseline knowledge (Guo et al., 2022). Since LLMs might be inconsistent on unknown facts, we consider a controlled setup of LLMs in retrieval-augmented generation (RAG) that references an authoritative knowledge base to an LLM beyond training data while generating a response (Lewis et al., 2020). As a RAG context, we resort to real-world knowledge graphs (KGs) (Agrawal et al., 2023; Yang et al., 2023b; Suchanek & Luu, 2023; Huang et al., 2024b), which store large-scale facts in a structured format of $\langle$subject, relation, object$\rangle$ triples. Indeed, a KG is a rich source of highly-curated facts on entities connected by relationships, and provides a test bed for logical consistency assessment. Therefore in our setup, we aim to provide a KG context and a propositional fact-checking query to an LLM and assess the consistency of responses under logical manipulation. However, such a logical fact-checking dataset is absent in the literature.

**Contributions:** We have three-fold contributions in the paper.

**1. Benchmark.** We address the lack of a logical fact-checking dataset (LFC) over KGs by introducing three datasets: **FreebaseLFC**, **NELLLFC**, and **WikiLFC** (§5). They are derived from their KG counterparts, Freebase (Bordes et al., 2013), NELL (Carlson et al., 2010), and WikiKG90Mv2 (Hu et al., 2021b), respectively. Existing KG datasets, while being widely used, are not suitable for testing an LLM's logical consistency in their native format. Therefore, we transform a $\langle$subject, relation, object$\rangle$ triplet from the KG into a suitable input format for an LLM as a (Fact, Context) pair, referred to as LLMQuery – Figure 1 shows an example of a fact, its corresponding KG context, and the LLMQuery from the FreebaseLFC dataset.

**2. Assessment of Logical Consistency.** We propose logical consistency measures of LLMs on propositional logic-based queries, which cover fact-checking queries over KGs using retrieval-augmented LLMs (§3). The queried facts involve one or more triplets connected by logical operators

such as negation ($\neg$), conjunction ($\wedge$), and disjunction ($\vee$). For consistency assessment, we consider a variety of scenarios, such as both simple and complex facts, diverse logic rules, first-order logic, and `LLMQuery` with and without KG contexts (§5). We demonstrate that existing LLMs such as Llama2-7B, Llama2-13B, and Gemma-2B, while being more accurate with KG contexts, are not consistent when considering logical equivalents or variants of these facts.

**3. Improvement of Logical Consistency.** We demonstrate how existing LLMs can be fine-tuned to improve their consistency via supervised fine-tuning (Radford et al., 2019; Devlin et al., 2018) (§4). We showcase that *pre-train, prompt, and predict* paradigm is often insufficient to improve the consistency of LLMs for complex fact-checking with KG contexts, thus we resort to *pre-train, fine-tune, and predict* approach. Although supervised fine-tuning is shown to be effective in other downstream tasks (Wei et al., 2021; Li & Liang, 2021), it has not yet been conducted for logical fact-checking queries over KGs. Furthermore, fine-tuning is feasible in our case due to the availability of abundant training data with ground-truth labels from the KG (§4.2). Additionally, we leverage parameter-efficient fine-tuning based on QLoRA (Quantized Low-Rank Adaptation) (Dettmers et al., 2024), thus improving the fine-tuning efficiency.

Experimental results show that the fine-tuned LLMs improve the consistency by 14% on average, while our optimization techniques scale both fine-tuning and inferencing to large KGs and complex fact-checking tasks. Our experiments also show that models fine-tuned on simple facts generalize well to more complex, unseen facts and rules (§5). This is possible due to complex facts and rules being decomposable into a collection of simple facts (Proposition 1)[2].

## 2  BACKGROUND

We provide preliminaries on knowledge graphs and propositional logic, which constitute the background for assessing logical consistency in LLMs.

**Knowledge Graph (KG).** A knowledge graph stores large-scale and real-world facts as a graph, denoted by $\mathcal{G} = (\mathcal{V}, \mathcal{R})$, where $v \in \mathcal{V}$ is an entity and $r \in \mathcal{R}$ is a relation. $r$ is a binary function $r : \mathcal{V} \times \mathcal{V} \in \{0, 1\}$ indicating whether the relation $r$ holds between a pair of entities or not. In the KG, such a binary relation indicates a directed edge between the pair of entities, formally $u \xrightarrow{r} v$ indicates $r(u, v) = 1$. A KG $\mathcal{G}$ can be represented by a set of facts or triplets $\mathcal{T} \subseteq \mathcal{V} \times \mathcal{R} \times \mathcal{V}$: For a triplet $T = (u, r, v) \in \mathcal{T}$ with $u, v \in \mathcal{V}$, we refer to $u$ as a subject entity and $v$ as an object entity.

**Propositional Logic Facts.** At the core of a KG, propositional logic facts constitute one or multiple triplets connected by logical operators, including conjunction ($\wedge$), disjunction ($\vee$), and negation ($\neg$). As per the DNF theorem (Davey & Priestley, 2002; Halbeisen & Krapf, 2020), any propositional logic fact $q$ can be expressed in the disjunctive normal form (DNF), which is a disjunction of conjunctive facts, wherein each conjunctive fact is a conjunction of atomic relation facts. Formally,

$$q = c_1 \vee c_2 \vee \cdots \vee c_n \text{ where } c_i = e_{i_1} \wedge e_{i_2} \wedge \cdots \wedge e_{i_m}$$

$\{c_i | 1 \leqslant i \leqslant n\}$ are conjunction facts. A conjunction fact $c_i$ comprises one or more atomic relation facts $c_i = \wedge_{j=1}^{i_m} e_{ij}$. An *atomic relation fact* $e_{ij}$ is a relation projection between a pair of entities. Formally,

$$e_{ij} = r(u, v) \text{ or } e_{ij} = \neg r(u, v)$$

where $u \in \mathcal{V}$ is a subject entity, $v \in \mathcal{V}$ is an object entity, and $r \in \mathcal{R}$ is a relation. We refer to an atomic relation fact as the *simple fact*, while a propositional logic fact involving multiple triplets connected by logical operators is called a *complex fact*.

---

[2]We analyze the impact of considering KG contexts (Appendix G.2), context length (Appendix G.3), different KG retrieval methods to build the KG context in `LLMQuery` (Appendix G.4), different prompting strategies (Appendix G.5), and the impact of learning rate (Appendix G.6) and epochs (Appendix G.7) on logical consistency and accuracy in fact-checking. We discuss the generalization of logical consistency to complex queries (Appendix H). We extend consistency assessment beyond structured format to natural text queries in Appendix I. Also, we demonstrate the superiority of our fact-checking framework over existing fact-checker in Appendix J. We experiment with Llama2-70B and GPT-4o, where instruction prompting tends to be sufficient to improve logical consistency since fine-tuning is often infeasible in large and closed-source models (Appendix K). Finally, we empirically verify whether LLMs fine-tuned to be logically consistent retain their performance on general language benchmarks in Appendix L.

Given a propositional logic fact $q$, let us denote its truth value as $\mathbf{T}(q) \in \{0,1\}$. The focus of this work is to find the truth value of a propositional logic fact, also referred to as the *fact-checking*, using an LLM and a KG context.

**Example 1.** $q_1 = \texttt{participantCountry}(2008\ \text{Olympics}, \text{Vatican City})$ is a simple fact. The truth value of $q_1$, denoted as $\mathbf{T}(q_1) = 0$, as Vatican City did not participate in the 2008 Olympics. Let us consider a complex fact $q_2 = \texttt{participantAthlete}(2020\ \text{Olympics}, \text{Nelly Korda})\ \wedge \texttt{awardedTo}(\text{Olympics Gold Medal}, \text{Nelly Korda})$. In this case, the truth value $\mathbf{T}(q_2) = 1$, since both of the constituent simple facts are true.

## 3 MEASURING LOGICAL CONSISTENCY ON PROPOSITIONAL LOGIC FACTS

In this section, we motivate and define the quantitative measure of logical consistency of an LLM response on propositional logic facts, presented as `LLMQuery` (Figure 1). When prompted the LLM with a fact $q$, the LLM either accepts $q$ as a valid fact or rejects it, that is, $\texttt{LLM}(q) \in \{0,1\}$, which is a binary function that either validates $q$ with response 1, or invalidates $q$ with response 0. In this paper, we guide the LLM to be a binary classifier by auto-regressively generating binary responses such as yes/no, accept/reject, etc., via prompt instructions. Next, we discuss the correctness criteria of $\texttt{LLM}(q)$ followed by consistency measures.

**Definition 1** (**Correctness of fact-checking via LLM response**). Given a fact $q$, an LLM response is correct if $\texttt{LLM}(q) = \mathbf{T}(q)$. In other words, a correct LLM response produces 1 for a true fact ($\mathbf{T}(q) = 1$) and 0 for a false fact ($\mathbf{T}(q) = 0$). The correctness can be extended to measure the average fact-checking accuracy of the LLM on a batch $Q = \{q_k : 1 \leqslant k \leqslant N\}$ of $N$ facts as $\texttt{Accuracy}(\texttt{LLM}, Q) = \frac{1}{N}\sum_{k=1}^{N} \mathbb{1}_{\texttt{LLM}(q_k)=\mathbf{T}(q_k)}$, where $\mathbb{1}$ is an indicator function taking value 1 if the LLM is correct for the fact $q_k$, and 0 otherwise.

Note that an LLM response being correct (or incorrect) says nothing about whether the response is consistent or not. We clarify this point with an example, where the LLM is correct in one out of two fact-checking queries, yet inconsistent as a whole.

**Example 2.** Consider a true fact $q = \texttt{honoredFor}(\text{Golden Globe Award for Best Director}, \text{Born on the Fourth of July})$ from Freebase KG and a false fact $\neg q = \texttt{honoredFor}(\text{Golden Globe Award for Best Director}, \text{Chamada a Cobrar})$. In both $q$ and $\neg q$, the subject entity is an award category and the object entity is a film. Next, we formulate an `LLMQuery` as a (Fact $q$, Context)-pair as in Figure 1 – we defer the construction of KG context to § 3.4, where the context intuitively contains the necessary knowledge for the LLM in validating $q$ or $\neg q$. Upon prompting Llama2-7B model with (Fact $q$, Context), we obtain the response $\texttt{LLM}(q) = 1$. On the contrary, when we prompt Llama2-7B with a negative `LLMQuery` (Fact $\neg q$, Context) with the same context but a negated fact, we receive an identical response $\texttt{LLM}(\neg q) = 1$. Therefore, the LLM responses are **inconsistent** as it returns 1 to both true and false facts and fails to distinguish between logically opposing facts.

It is well-known that the set of operators $\{\neg, \wedge, \vee\}$ is functionally complete (Howson, 2005), implying that any propositional logic statement can be expressed using these three operators. Hence, we start with defining an LLM response's consistency for these three operators.

**Definition 2** (**Logical Consistency on Primitive Operators**). Let us consider $p$ and $q$ as propositional logic facts. An LLM response is logically consistent if the following conditions are satisfied.

$$\texttt{LLM}(\neg q) = \neg\texttt{LLM}(q) \tag{1}$$

$$\texttt{LLM}(p \vee q) = \texttt{LLM}(p) \vee \texttt{LLM}(q) \tag{2}$$

$$\texttt{LLM}(p \wedge q) = \texttt{LLM}(p) \wedge \texttt{LLM}(q) \tag{3}$$

Informally, *the consistency criteria states that LLM responses on constituent sub-queries must comply with the semantics of the logical operators connecting those sub-queries.*

Below, we discuss the LLM's consistency w.r.t. these three operators in detail with KG contexts.

### 3.1 CONSISTENCY OF SIMPLE FACTS

A simple fact is an atomic relation; therefore, consistency on negation is applicable (Equation 1). As stated in §2.1, a simple fact in a KG is either a relation $r(u,v)$ or its negation $\neg r(u,v)$. The negation

$\neg r(u, v)$ can be computed in two ways: **(1)** by replacing $r$ in the triplet $(u, r, v)$ with an alternative falsified relation $r'$ s.t. $r'(u, v) = 0$; or **(2)** via negating entities, e.g., by finding an alternate object entity $v'$ such that $r(u, v') = 0$. We adopt the second approach for practical purposes since the notion of consistency does not depend on how the negation of a triplet is computed.

Given a simple fact $q = r(u, v)$, we check consistency with respect to the negation operator. First, we compute the negation of this fact, $\neg q = r(u, v')$ by finding an alternate object entity $v'$, and finally, we check the consistency of LLM responses based on whether the following holds (equation 1).

$$\text{LLM}(r(u, v')) = \neg\text{LLM}(r(u, v))$$

Example 2 showed an example of logical inconsistency. In the following, we show an example of logical consistency of a simple fact.

**Example 3.** Consider a true fact $q = \texttt{capital}(\text{Vietnam, Hanoi})$ from Freebase and a false fact $\neg q = \texttt{capital}(\text{Vietnam, Sorkh Qaleh - North Khorasan})$. We formulate an LLMQuery as a (Fact $q$, Context)-pair similar to that in Figure 1. Upon prompting Llama2-7B, we obtain the response $\text{LLM}(q) = 1$. On a similarly constructed negation LLMQuery = (Fact $\neg q$, Context), we find the response $\text{LLM}(\neg q) = 0$. Since $\text{LLM}(\neg q) = 0 = \neg 1 = \neg\text{LLM}(q)$, Llama2-7B is logically **consistent**.

Similar to accuracy, we measure the average consistency of LLM responses on a given batch containing $N$ pairs of simple facts and negations, $Q = \{(q_k, \neg q_k) : 1 \leq k \leq N\}$: $\text{Consistency}(\text{LLM}, Q) = \frac{1}{N} \sum_{k=1}^{N} \mathbb{1}_{\text{LLM}(q_k) = \neg\text{LLM}(\neg q_k)}$.

### 3.2 CONSISTENCY OF COMPLEX FACTS

The canonical example of a complex fact is a DNF fact since a DNF fact consists of functionally complete logical operators $\{\neg, \wedge, \vee\}$. Indeed, one can construct complex facts using other operators, such as implication ($\Rightarrow$) and biconditional ($\Leftrightarrow$), however, such facts can be transformed into DNF (Davey & Priestley, 2002). Therefore, we define the consistency of a complex fact as the consistency of a DNF fact and apply similar constraints as in Definition 2: Informally, an LLM is consistent to a DNF fact if the LLM response on the DNF fact is equal to applying a hierarchy of disjunctions followed by conjunctions on the LLM responses over the lowest level atomic facts.

**Proposition 1.** An LLM is consistent on a DNF fact $q = \vee_{i=1}^{n} c_i$, where $c_i = \wedge_{j=1}^{i_m} e_{ij}$, if $\text{LLM}(q) = \bigvee_{i=1}^{n} \left( \bigwedge_{j=1}^{i_m} \text{LLM}(e_{ij}) \right)$. Here, $e_{ij}$ is an atomic relation fact for any $1 \leq i \leq n$ and $1 \leq j \leq i_m$.

The proof is given in the Appendix C. Since any propositional logic fact can be expressed in the DNF form, Proposition 1 encompasses the consistency criteria of any propositional logic fact. In addition, we measure the average consistency given a batch of $N$ DNF queries $Q = \left\{ q_k = \vee_{i=1}^{n} \left( \wedge_{j=1}^{i_m} e_{ij}^{k} \right) : 1 \leq k \leq N \right\}$ as $\text{Consistency}(\text{LLM}, Q) = \frac{1}{N} \sum_{k=1}^{N} \mathbb{1}_{\text{LLM}(q_k) = \vee_{i=1}^{n} \left( \wedge_{j=1}^{i_m} \text{LLM}(e_{ij}^{k}) \right)}$.

The choice of DNF in Proposition 1 is not the only design choice, since the proposed method for assessing logical consistency is not reliant on any specific normal form. In fact, the definition of consistency is flexible enough to be adapted to other canonical forms such as CNF (Proposition 3). In the following, as a basic example of complex facts, we show an LLM's consistency assessment using conjunction with two atomic relation facts from Freebase.

**Example 4.** Consider two true facts from Freebase: $p = \texttt{filmProduction}(\text{Universal Studios, The Family Man})$ and $q = \texttt{filmProduction}(\text{Relativity Media, Valentine's Day})$. By prompting Llama2-7B with an LLMQuery constituting a complex fact with $\wedge$ operator, i.e., the (Fact $p \wedge$ Fact $q$, Context)-pair, we obtain the response $\text{LLM}(p \wedge q) = 1$. Similarly, by prompting the (Fact $p$, Context)-pair, we receive the response $\text{LLM}(p) = 1$. We obtain the same for q, $\text{LLM}(q) = 1$. In this case, the LLM is **consistent** because $\text{LLM}(p \wedge q) = \text{LLM}(p) \wedge \text{LLM}(q)$.

Consider the true fact $p = \texttt{countryReverse}(\text{Brazil, Artistic gymnastics})$ implying that Artistic Gymnastic is played in Brazil and another true fact $q = \texttt{sport}(\text{2012 Summer Olympics, Artistic gymnastics})$. Upon prompting Llama2-7B with an LLMQuery with (Fact $p \wedge$ Fact $q$ , Context)-pair, we obtain the response $\text{LLM}(p \wedge q) = 1$. However, upon prompting Llama2-7B with the constituent simple facts individually, we receive the response $\text{LLM}(p) = 0$ and $\text{LLM}(q) = 1$. In this case, the LLM is **inconsistent** because $\text{LLM}(p \wedge q) = 1 \neq 0 = \text{LLM}(p) \wedge \text{LLM}(q)$.

---

**Algorithm 1** LLM fact-checking with KG contexts (for simple facts)

---

    **Input:** A language model LLM, a knowledge graph $\mathcal{G}$, a simple fact $r(u, v)$, maximum hop $\lambda$
    **Output**: A binary value $\{0, 1\}$
1:  $G_{\text{sub}} \leftarrow \texttt{BoundedBFS}(\mathcal{G}, u, \lambda)$             ▷ Find the $\lambda$ hop subgraph of the subject $u$ in $\mathcal{G}$
2:  $\varphi \leftarrow \texttt{BuildContext}(G_{\text{sub}}, r)$           ▷ Get the relevant context for relation $r$ in $G_{\text{sub}}$
3:  **return** $\texttt{LLMQuery}(r(u,v), \varphi)$  ▷ Get a binary response on concatenated context and target fact

---

### 3.3   CONSISTENCY FOR LOGIC RULES

An important family of true facts comes from various logical rules of inference, such as the commutative, associative, and distributive properties. One critical question is: *Do LLMs respond such that these laws of inference are respected?* Let $p, q, s$ be propositional logic facts. An LLM is consistent w.r.t. the commutative law if reordering logic facts results in an identical response.

$$\texttt{LLM}(p \vee q) = \texttt{LLM}(q \vee p) \text{ and } \texttt{LLM}(p \wedge q) = \texttt{LLM}(q \wedge p)$$

An LLM is consistent w.r.t. the associative law if the following holds.

$$\texttt{LLM}((p \vee q) \vee s) = \texttt{LLM}(p \vee (q \vee s)) \text{ and } \texttt{LLM}((p \wedge q) \wedge s) = \texttt{LLM}((p \wedge (q \wedge s))$$

An LLM is consistent w.r.t. the distributive law if the following holds.

$$\texttt{LLM}(p \wedge (q \vee s)) = \texttt{LLM}((p \wedge q) \vee (p \vee s)) \text{ and } \texttt{LLM}(p \vee (q \wedge s)) = \texttt{LLM}((p \vee q) \wedge (p \vee s))$$

Given a batch of $N$ logic rules $L = \{l_k : 1 \leqslant k \leqslant N\}$, we measure the average consistency of LLM responses as $\texttt{Consistency}(\texttt{LLM}, L) = \frac{1}{N} \sum_{k=1}^{N} \mathbb{1}_{\texttt{Consistency}(\texttt{LLM}, l_k)}$, where $\mathbb{1}_{\texttt{Consistency}(\texttt{LLM}, l_k)}$ is an indicator function taking value 1 if the LLM is consistent for the logic rule $l_k$, and 0 otherwise.

### 3.4   CONSTRUCTING LLMQuery WITH FACTS AND KG CONTEXTS

Our goal is to assess the logical consistency of LLMs on fact-checking queries from KGs, as discussed above. It is plausible that LLMs lack the necessary knowledge to answer the fact-checking query accurately, let alone being consistent, especially on recent facts never seen during pre-training/fine-tuning. As such, motivated by the RAG-based framework, we intend to provide necessary knowledge or *context* from the KG in LLMQuery before assessing for consistency. Algorithm 1 provides a template pipeline for LLMQuery, where a key step is to find the relevant context for the target fact. Among different choices, we first discuss a graph traversal-based method for context retrieval, followed by alternate methods such as vector-embedding-based retrieval.

**Graph Traversal.** Given a target fact, we apply breadth-first search (BFS) to get a relevant subgraph from the KG. For simplicity, let us consider a simple fact either in the form $r(u, v)$, or in the negation form $\neg r(u, v)$. The relevant information of an entity is generally expected to be in the local neighbourhood in the KG (Dettmers et al., 2018; Wang et al., 2020b). Hence, we conduct BFS from the subject entity $u$ up to a certain number of hops (e.g., $\lambda = 2$ or 3) in the KG, and get a connected subgraph (line 1). To analyse the complexity of BFS, let the $\lambda$-hop neighbourhood of $u$ contain $|V_\lambda(u)|$ nodes and $|E_\lambda(u)|$ edges in the KG. The time complexity of BFS-based context building is $O(|V_\lambda(u)| + |E_\lambda(u)|)$. Thereafter, we provide the list of triplets in the subgraph as a context to LLMQuery (line 2). Our hypothesis is that the context contains the necessary information for the LLM to validate the target fact: If the fact is implied by the context, the LLM is expected to return the binary response 1 (validating the fact), and 0 otherwise (invalidating it).

**Optimizing Context Generation.** LLMs are restricted to processing a finite-length context. Besides, LLMs' performance often deteriorates with long contexts (Liu et al., 2023a). Therefore, we further prune the context/subgraph from BFS, especially when the subject entity $u$ is connected to many other entities within a few hops. We adopt three potential approaches: (**1**) We prioritize triplets having semantically similar relations as the target fact's relation $r$. We identify such semantically similar relations by utilizing the ontology and relation hierarchy, often available with KG (Chah, 2018; Chen et al., 2023). For a complex fact, we split the context size equally for each simple fact involved in it. (**2**) We apply vector-embedding-based methods to project triplets in the subgraph to a high-dimensional embedding space (Reimers & Gurevych, 2019) and find relevant triplets

> Consider the context: $Context.
> Is the fact correct given the context? Answer Yes when the fact is in the context and No otherwise. $Fact.

Figure 2: The adaptation of LLMQuery from Figure 1 for instruction prompting. The prompt contains a clear instruction (in blue) to guide the LLM towards outputting a correct response.

close to the embedded target fact using approximate nearest neighbor (ANN) algorithms (Malkov & Yashunin, 2020). (**3**) As the last approach, we bypass applying BFS, embed the entire KG triplets, and retrieve relevant triplets to the target fact. Thus, we provide the most relevant context to the LLM so that we can precisely assess its logical consistency. Furthermore, our framework for assessing consistency is modular and can benefit from future development in graph-based retrieval of related contexts such as graphRAG (Liu et al., 2024; Sanmartin, 2024).

## 4    IMPROVING THE LLM CONSISTENCY BY SUPERVISED FINE-TUNING

Supervised fine-tuning is the process of fine-tuning LLMs on a specific task by providing them with explicit instructions or prompts along with examples of inputs (training data) and outputs (ground-truth labels) relevant to that task. These prompts guide the model's learning process and help it specialize in that task. The task considered in this paper is logically consistent fact-checking. Fine-tuning is feasible for this task because there are abundant facts available in the KG for fine-tuning. We first discuss the reasons behind adopting supervised fine-tuning rather than zero-shot instruction prompting for this task. Afterwards, we discuss the steps involved in the fine-tuning.

### 4.1    MOTIVATION: INSTRUCTION PROMPTING VS. SUPERVISED FINE-TUNING

Supervised fine-tuning requires time and computational resources, and in some cases, it is not needed at all (Radford et al., 2019; Ziegler et al., 2019). Hence, it is important to verify if our task necessitates fine-tuning. We first consider zero-shot instruction prompting – shown in Figure 2, as a computationally cheaper alternative. Our goal is to instruct the LLM to output yes when the fact triplets are inside the context, and no otherwise. The context length is limited to 1000 tokens. We empirically find that even for simple facts checking, such zero-shot instruction prompting does not improve the logical consistency, while supervised fine-tuning outperforms zero-shot instruction prompting; the consistency improves by up to 46% (see Tables 3 and 11).

The reasons for inferior performance of zero-shot instruction prompting compared to supervised fine-tuning are two-fold: (1) *Nuances and intricacies of logical facts.* It is well known that supervised fine-tuning of pre-trained LLMs is preferable for tasks that require precise and nuanced predictions, especially when the task involves multiple classes or fine-grained distinctions (Wei et al., 2021; Li & Liang, 2021). In such cases, providing explicit supervision through labeled data allows the LLM to adjust its behaviour based on the nuances and intricacies of the task domain and learn to make accurate predictions based on the task-specific examples (Sainz et al., 2024). For a simple fact-checking task, the intricacies arise from understanding the differences between a true fact and its corresponding false fact. (2) *Limited knowledge.* Fine-tuning a pre-trained LLM with task-specific labeled examples enables it to directly optimize its parameters for the target task, leading to potentially better performance compared to zero-shot instruction prompting (Ziegler et al., 2019; Xu et al., 2023), which relies on generalization from instructions given in the prompt. For instance, supervised fine-tuning has shown better performance than zero-shot prompting on tasks such as named entity recognition, relation extraction, and relation classification (Sainz et al., 2024; Xu et al., 2023), while Ziegler et al. (2019) showed that supervised fine-tuning with explicit task-specific instructions or preferences performs better than zero-shot prompting on text summarization task. In the case of our simple fact-checking task, without sufficient data, the LLM fails to learn significantly different representations of a false fact from a true fact.

### 4.2    METHODOLOGY FOR SUPERVISED FINE-TUNING

We consider supervised fine-tuning to align LLM responses to be logically consistent. We consider a two-step procedure: dataset creation and supervised fine-tuning. Here, we discuss dataset creation and refer to Appendix D for details on fine-tuning and the two-step Algorithm 2.

**Instruction Dataset Creation.** We construct a binary instruction dataset from primitive logic facts containing negation, conjunction, and disjunction operators – the goal is to help the LLM *correctly classify* logic facts as true/false. More specifically, we construct a dataset with four facts: $p, \neg p, p \vee q$, and $p \wedge q$ preceded by a context $\varphi$. For simple facts, the ground truth response of $p$ is true when $p \in \varphi$ and false otherwise – the LLM is expected to learn a function where if the target fact is in the context, it should recognize this and outputs $1$, and $0$ otherwise. For complex facts, the ground truth response for $p \vee q$ is true when $p \in \varphi$ or $q \in \varphi$, and false otherwise. Similarly, the response for $p \wedge q$ is true when both $p \in \varphi$ and $q \in \varphi$. Assume a simple fact $p$ having subject node $u$, and the $\lambda$-hop neighbourhood of $u$ contains $|V_\lambda(u)|$ nodes and $|E_\lambda(u)|$ edges in the KG. Following §3.4, the time complexity of creating instruction data for the simple fact $p$ is $O(|V_\lambda(u)| + |E_\lambda(u)|)$. In addition, we prioritize keeping both positive and negative training examples in the same batch so that the LLM learns to distinguish between them and subsequently becomes more consistent. Below, we formalize the sufficient condition to achieve logical consistency by achieving accuracy in fact-checking.

**Proposition 2.** *(I) An LLM is consistent on a simple atomic fact if it is accurate both on the fact and its negation. (II) For a complex DNF fact, the LLM is consistent if it is accurate on the DNF fact as well as on all constituent atomic facts.*

We explain Proposition 2 using examples and defer the proof to the Appendix C. Consider a simple fact $p$, where $p \in \varphi$. Let the LLM accurately classify $p$ as $1$ and $\neg p$ as $0$. Then, the LLM is consistent on $p$. Further, consider a conjunction fact $p \wedge q$, where $p \in \varphi$ and $q \notin \varphi$. Let the LLM accurately classify $p \wedge q$ as $0$, and constituent atomic facts $p$ as $1$ and $q$ as $0$. Therefore, the LLM is consistent on $p \wedge q$. Notice that Proposition 2 provides a sufficient (but not necessary) condition for the LLM's logical consistency on both simple and complex facts. Therefore, we expect that fine-tuning to enhance the accuracy following Proposition 2 would lead to the LLM's logical consistency.

**Generalization to Complex Facts and Logic Rules.** Our instruction dataset focuses on accurately classifying single logic operators such as negation, conjunction, and disjunction. Recognizing primitive operators is the building block for classifying facts and rules with multiple heterogeneous operators. Our hypothesis therefore is that fine-tuning to recognize single operators should generalize to complex logic facts and rules with multiple operators, which we informally attribute as the emergent abilities of LLMs (Lu et al., 2023; Schaeffer et al., 2024). In this work, we empirically showcase the generalization to out-of-distribution complex facts and logic rules in § 5.

## 5 EXPERIMENTAL RESULTS

We conduct experiments to evaluate the logical consistency of LLMs and the impact of supervised fine-tuning on improving consistency. Additional details on experiments, results, and code are in the supplementary materials. Specifically, we investigate the following research questions empirically.

- **Logical Consistency via Improved Accuracy.** Does the logical consistency of LLMs improve via a fine-tuning task of correctly classifying primitive logical operators?
- **Generalizability.** Does the fine-tuned models generalize to in-distribution facts from different datasets, and more generally, to out-of-distribution logic facts containing more operators?
- **Efficiency.** How is the efficiency of PEFT vs. full fine-tuning and LLM inference on logic facts?
- **Ablation Study.** Does instruction prompting improve logical consistency compared to fine-tuning across different model-sizes? What is the impact of different KG retrieval methods on the consistency of the LLM? Does adding KG contexts help LLMs in improving accuracy and consistency, and how does the length of context play a role? What is the impact of hyperparameters, e.g., learning rate in fine-tuning?

### 5.1 DATASETS AND EXPERIMENTAL SETUP

**KG Datasets and Facts.** We consider three KG benchmarks: Freebase (FB15K), NELL, and a large-scale dataset from OGB: WikiKG90Mv2 (Wiki) (Hu et al., 2021b). We obtain FB15K and NELL from the codebase of Query2Box (Ren et al., 2020). We create logical fact-checking datasets FreebaseLFC, NELLLFC, and WikiLFC using the FB15K, NELL, and Wiki KGs, respectively. Statistics of these KGs and fact-checking datasets are in Tables 4-5 in the Appendix G. In all experiments, we resort to our proposed fact-checking datasets, since – to our best knowledge – none of the earlier works focus on assessing and improving LLM's logical consistency in fact-checking.

Table 1: Accuracy and logical consistency of LLMs before and after fine-tuning (FT). Bold numbers denote an improved result in accuracy and consistency for fine-tuning. Training is performed on FreebaseLFC dataset only (marked with '∗'), while performance improved in all datasets.

| Model | Dataset | Fact | Accuracy | | Logical Consistency | |
|---|---|---|---|---|---|---|
| | | | Before FT | After FT | Before FT | After FT |
| Llama2-13B | FreebaseLFC∗ | $p, \neg p$ | 0.90 | **0.93** | 0.81 | **0.86** |
| | | $p \wedge q$ | 0.61 | **0.93** | 0.67 | **0.83** |
| | | $p \vee q$ | 0.73 | **0.76** | 0.73 | **0.97** |
| | NELLLFC | $p, \neg p$ | 0.88 | **0.97** | 0.76 | **0.93** |
| | | $p \wedge q$ | 0.38 | **0.89** | 0.69 | **0.88** |
| | | $p \vee q$ | 0.73 | **0.76** | 0.73 | **0.94** |
| | WikiLFC | $p, \neg p$ | 0.96 | **0.96** | 0.92 | **0.93** |
| Llama2-7B | FreebaseLFC∗ | $p, \neg p$ | 0.87 | **0.97** | 0.76 | **0.94** |
| | | $p \wedge q$ | 0.39 | **0.87** | 0.47 | **0.86** |
| | | $p \vee q$ | 0.78 | **0.81** | 0.83 | 0.77 |
| | NELLLFC | $p, \neg p$ | 0.71 | **0.94** | 0.44 | **0.88** |
| | | $p \wedge q$ | 0.26 | **0.81** | 0.71 | **0.91** |
| | | $p \vee q$ | 0.75 | 0.73 | 0.95 | 0.78 |
| | WikiLFC | $p, \neg p$ | 0.90 | **0.90** | 0.80 | **0.81** |
| Gemma-2B | FreebaseLFC∗ | $p, \neg p$ | 0.82 | **0.98** | 0.66 | **0.96** |
| | | $p \wedge q$ | 0.45 | **0.83** | 0.70 | **0.84** |
| | | $p \vee q$ | 0.73 | **0.94** | 0.91 | 0.90 |
| | NELLLFC | $p, \neg p$ | 0.81 | **0.94** | 0.62 | **0.89** |
| | | $p \wedge q$ | 0.33 | **0.83** | 0.78 | **0.83** |
| | | $p \vee q$ | 0.75 | **0.88** | 0.98 | 0.87 |
| | WikiLFC | $p, \neg p$ | 0.76 | **0.90** | 0.52 | **0.80** |
| Average | | | 0.69 | **0.88** | 0.74 | **0.88** |

**LLMs.** We evaluate the logical consistency of three open-source LLMs: the chat version of Llama2-7B (Touvron et al., 2023) and Llama2-13B (Touvron et al., 2023) with 7B and 13B parameters, respectively, and instruction tuned Gemma-2B with 2B parameters (Team et al., 2024). We fine-tune 20 epochs and save the intermediate models. In order to select the best model for each dataset and fact type, we consider the following principle: we compute the sum of accuracy and consistency on both evaluation and test set (to prioritize both accuracy and consistency), find the optimal epoch having the maximum sum on the evaluation set, and report results on respective test set.

## 5.2 EXPERIMENTAL RESULTS

**Improved Logical Consistency via Improved Accuracy.** Table 1 shows the result for simple facts $(p, \neg p)$ and facts with one logical operator. In all LLMs and FreebaseLFC dataset, fine-tuning results in higher accuracy and consistency in most fact types – the exception is for disjunctive fact $p \vee q$ where fine-tuning demonstrates a higher accuracy yet a slight decrease in consistency (further explanation in Appendix G.3). We observe a similar trend in NELL and Wiki datasets – which are not included in fine-tuning – revealing the generalization effect of fine-tuning on similar facts in different datasets. Within Llama2 base models (before fine-tuning), the 13B model has, in general, higher accuracy and consistency on logic facts than the 7B model – thus, more model parameters achieve improved accuracy and consistency, which is further improved via fine-tuning. *Therefore, fine-tuning logic facts for increasing accuracy in one dataset increases accuracy (on average* $19\%$*) and logical consistency (on average* $14\%$*) across all datasets.*

**Generalizability of Fine-tuned Models.** Table 2 shows the generalizability of fine-tuned models, i.e., increase in accuracy and consistency across facts with more operators and commutative rules. In particular, while accuracy is always higher on out-of-distribution facts, there is an occasional

Table 2: Generalization of fine-tuning to complex facts and rules in Llama2-13B. For results on Llama2-7B and Gemma-2B, refer to Table 13 in the Appendix G

| Model | Dataset | Fact | Accuracy | | Logical Consistency | |
|---|---|---|---|---|---|---|
| | | | Before FT | After FT | Before FT | After FT |
| Llama2-13B | FreebaseLFC | $p \vee (q \wedge r)$ | 0.74 | **0.85** | 0.78 | **0.81** |
| | | $p \wedge (q \vee r)$ | 0.53 | **0.82** | 0.63 | **0.79** |
| | | $p \vee q \leftrightarrow q \vee p$ | 0.73 | **0.76** | 0.85 | **0.98** |
| | | $p \wedge q \leftrightarrow q \wedge p$ | 0.61 | **0.93** | 0.84 | **0.91** |
| | NELLLFC | $p \vee (q \wedge r)$ | 0.67 | **0.77** | 0.72 | **0.80** |
| | | $p \wedge (q \vee r)$ | 0.42 | 0.38 | 0.75 | **0.85** |
| | | $p \vee q \leftrightarrow q \vee p$ | 0.73 | **0.76** | 0.80 | **0.98** |
| | | $p \wedge q \leftrightarrow q \wedge p$ | 0.38 | **0.70** | 0.90 | 0.85 |

Table 3: Impact of instruction prompting (Prompt) vs. fine-tuning (FT) on simple facts. Fine-tuning results in higher accuracy and consistency than instruction prompting. Extended result is in Table 11.

| Model | Dataset | Accuracy | | | Logical Consistency | | |
|---|---|---|---|---|---|---|---|
| | | Before FT | Prompt | After FT | Before FT | Prompt | After FT |
| Llama2-13B | FreebaseLFC | 0.90 | 0.89 | **0.93** | 0.81 | 0.79 | **0.86** |
| | NELLLFC | 0.88 | 0.85 | **0.97** | 0.76 | 0.72 | **0.93** |
| | WikiLFC | 0.96 | 0.94 | **0.96** | 0.92 | 0.90 | **0.93** |

decrease in consistency, specially in facts containing a disjunctive operator. *Therefore, nudging the LLMs to recognize primitive operators can generalize to complex facts and rules.*

**Efficiency.** Our empirical results demonstrate that *PEFT is 3X more efficient than full fine-tuning for logical consistency. The end-to-end fact-checking time is also efficient (less than 0.63 seconds per sample) in our datasets.* For details, we refer to Tables 6-7 in the Appendix G.

**Ablation Study: Supervised Fine-tuning vs. Zero-shot Instruction Prompting.** Table 3 shows the effect of zero-shot instruction prompting (an example in Figure 2) vs. fine-tuning on the accuracy and logical consistency of simple facts. First, instruction prompting negatively impacts the LLM reasoning on Llama2 base models (before fine-tuning), where both accuracy and consistency is lower compared to not considering any instruction. In contrast, fine-tuning results in higher accuracy and logical consistency than instruction prompting in all datasets and fact types, *thereby validating the need for fine-tuning to improve accuracy and logical consistency of logic facts.*

## 6 CONCLUSIONS

We propose a methodology to assess and improve the logical consistency of LLMs in fact-checking from KGs. We formalize the definition of logical consistency of LLMs on propositional logic queries comprising multiple logical operators. LLMs are usually less consistent in answering logic facts. A supervised fine-tuning for recognizing primitive logic operators (negation, conjunction, and disjunction) improves their logical consistency – we show a $14\%$ increase in consistency via fine-tuning.

Our work has potential impacts on trustworthy language modeling. A logically consistent LLM is less likely to hallucinate since its response is unlikely to change due to the logical manipulation of the prompt. Future works can be in several directions. Beyond binary responses, we aim to assess consistency when LLM responses are not trivially classified into finite categories. We will extend consistency assessment to natural language queries containing logical relations. We will consider unstructured context unlike this paper, which further complicates LLM's ability to accurately and consistently answer fact-checking queries. On the KG front, we investigate the efficacy of knowledge injection methods, or graph-to-text conversion, to input more relevant contexts in the LLMQuery. Finally, incorporating link prediction and reasoning capabilities into LLMs to conduct logic fact-checking over incomplete KGs is an interesting problem.

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

## A    LIMITATIONS AND BROADER IMPACTS

Our accuracy and consistency measures are given over logic facts for which ground-truths are available from the input KG and LLM responses are binary. In this work, we do not consider link prediction and multi-hop reasoning capabilities of LLM models to validate facts over incomplete KGs. It is vital for both researchers and investigators to exercise prudent judgment and validate findings through additional comparative methods before arriving at any definitive conclusions or undertaking consequential actions.

LLM's general reasoning ability has been under much scrutiny in recent years, and experiments suggest that there is still a gap between LLMs and human-like logical reasoning ability. Given such a state of affairs, one limitation of our current study is that we do not evaluate how much logical consistency improves the general reasoning ability of LLMs. Another limitation is that our paper mainly focuses on binary responses, and does not consider extensions to multi-class logic and non-binary answer consistency. We will consider such extensions in the future.

## B    RELATED WORKS

### B.1    LARGE LANGUAGE MODELS WITH KG CONTEXTS

Large language models (LLMs), pre-trained on large-scale web and enterprises corpus, encode significant knowledge implicitly in its parameters without human supervision, which can be probed for various question-answering and querying tasks, thus LLMs act as knowledge bases (Petroni et al., 2019; Hao et al., 2023; Wang et al., 2020a). LLMs have also shown tremendous promise in new tasks for which they have not been pre-trained. Users can interact with state-of-the-art LLMs through prompting. They simply specify instructions for new tasks as text prompts to guide an LLM for the desired outputs (Liu et al., 2023b). However, LLMs are proficient at learning probabilistic language patterns and may not explicitly store consistent representations of knowledge, hence they can output unreliable and incoherent responses and often experience hallucinations by generating factually incorrect statements (Elazar et al., 2021; Liu et al., 2023c; Sun et al., 2024a).

Knowledge graphs (KGs) can enhance pre-training (Yu et al., 2022), fine-tuning (Wang et al., 2024b), inference (Wei et al., 2023), prompting (Baek et al., 2023), retrieval/ knowledge augmented generation (Xu et al., 2024), knowledge editing (Zheng et al., 2023), and knowledge validation (Hu et al., 2024a) of LLMs. In particular, KGs can assist in the inference phase of LLMs through retrieval-augmented generation (RAG), e.g., to provide KG context within prompts for reducing hallucinations, thus improving accuracy and consistency. For instance, Liu et al. (2020) inject KG triplets into LLMs to enhance the domain knowledge, meanwhile Li et al. (2023) uses external data sources to improve the factual correctness of large language models. GraphRAG is the latest technology that models an external knowledge base as a knowledge graph to improve retrieval methods with a more comprehensive contextual understanding and thereby assists in obtaining more precise search results via LLMs (Xu et al., 2024; Edge et al., 2024; Sanmartin, 2024; Liu et al., 2024). Since we consider a RAG paradigm for LLM's consistency assessment, a good retrieval technique (e.g., future developments on graphRAG) can be leveraged in our framework to further improve the LLM's accuracy and consistency in fact-checking.

### B.2    MEASURING AND IMPROVING THE CONSISTENCY OF LLMS

With the prevalence of large language models, measuring their consistency has become critical. LLMs can generate logically contradicting outputs, e.g., different predictions for meaning-preserving text alternations (Ravichander et al., 2020; Elazar et al., 2021). They may violate important relational properties such as negation (Kassner & Schütze, 2020; Ettinger, 2020; Asai & Hajishirzi, 2020; Jang et al., 2022), symmetry (Wang et al., 2019; Li et al., 2019; Kumar & Joshi, 2022), and transitivity (Asai & Hajishirzi, 2020; Lin & Ng, 2022). Raj et al. (2023) measures the semantic similarity of LLM outputs in response to paraphrased versions of a question.

To improve the consistency of an LLM, several techniques are developed, e.g., adding a special consistency loss function to the original loss that penalizes inconsistent answers and continuing the pre-training step (Elazar et al., 2021). Others achieve this through data augmentation (Ray et al.,

2019; Asai & Hajishirzi, 2020). Some other works learn inter-relationships between concepts using external dictionaries (Jang & Lukasiewicz, 2023). There are also general approaches to improve an LLM's accuracy, but the same methods could be applied to enhance consistency, such as reinforcement learning from human feedback (Ouyang et al., 2022), prompt design optimization (Marvin et al., 2023), as well as model fine-tuning.

**First**, unlike the bulk of the literature that measures an LLM's consistency on natural language processing tasks, we develop systematic consistency measurement techniques of retrieval-augmented LLMs for propositional logic facts and logic rules, with knowledge graph contexts. **Second**, existing approaches generally update all model parameters and are inefficient. We avoid full fine-tuning by using parameter-efficient supervised fine-tuning through low-rank adaption. We have experimentally demonstrated that our approach is effective, efficient, and generalizes to complex facts and logic rules, can handle large-scale KGs, and LLMs having billions of parameters.

### B.3 LARGE LANGUAGE MODELS HALLUCINATIONS

LLMs may not explicitly store a consistent representation of knowledge. Hence, they can hallucinate by generating unreliable and incoherent responses, and often plausible yet factually incorrect statements. There are several types of LLM hallucinations. In this work, our focus is *faithfulness hallucination* (Huang et al., 2024a), resulting in divergence of LLM-generated contents from user inputs (e.g., KG contexts, instructions, examples, etc.), as well as the lack of consistency within the generated contents when the input queries are logically perturbed. Subsequently, various strategies have been developed to detect and reduce LLMs' hallucinations. For instance, retrieval-augmented generation (RAG) enhances models with up-to-date knowledge from external sources. More advanced approaches (Wang et al., 2024a) retrieve high-quality candidates to avoid the context poisoning issue, e.g., LLM-Embedder (Lv et al., 2024) is a general-purpose retrieval technique that learns vector embedding by rewarding high-quality retrieval candidates, contrastive learning, and knowledge distillation. In fact, we have already shown that vector embedding methods improve logical consistency over traversal-based context retrieval (Appendix E, Table 11). In the future, it would be interesting to replace the Sentence Transformer in our vector embedding method via LLM-Embedder to construct more relevant contexts. Another approach to reduce LLMs' hallucinations is the chain-of-thought (CoT) prompting (Wei et al., 2022), which encourages LLMs to generate reasoning steps before responding. We have also conducted experiments with CoT (Appendix § G.5.3), but it does not demonstrate any improvement over fine-tuning. As such, we hypothesize that achieving logical consistency requires an update in the internal weights of the LLMs, e.g., via supervised fine-tuning. Moreover, self-consistency (Wang et al., 2023a) employs a post-processing approach that does not improve the internal weights of the model and instead addresses the decoding module of the model – it samples multiple responses of the LLM and chooses a response with the majority vote that is denoted as consistent. In contrast, our fine-tuning procedure improves the internal weights of the LLM to be a consistent responder under logical perturbation of input queries, thereby reducing faithfulness hallucination.

### B.4 FACT-CHECKING WITH KGS AND LLMS

Knowledge graphs that store high-quality facts are reliable and valuable sources for fact-checking. We refer to Luo & Long (2021); Huynh & Papotti (2018) for surveys and experimental benchmarks. In particular, KG approaches for fact-checking can be broadly classified into paths (Shi & Weninger, 2016), rules and patterns (Lin et al., 2019; Galárraga et al., 2015), and embedding-based (Ammar & Celebi, 2019; Borrego et al., 2021). Recent embedding methods determine whether a given fact, currently missing in an incomplete knowledge graph, should appear in it. In contrast, our focus is to validate logic facts from the input KG.

LLMs like GPT-3.5 and GPT-4 have also been exploited for automated fact-checking with and without access to external contexts, e.g., Google search results (Quelle & Bovet, 2024). Hamed et al. (2023) studied fact-checking in biological graphs constructed from ChatGPT contents. Recently, MiniCheck (Tang et al., 2024) and RefChecker (Hu et al., 2024b) perform fact-checking of LLM outputs, primarily simple facts or *claim-triplets*, on grounding documents. To the best of our knowledge, ours is the first work on measuring and improving the consistency of LLMs for propositional logic-based fact-checking with KG contexts.

## C  FORMAL PROOFS

**Proposition 1.** An LLM is consistent with respect to a DNF fact $q = \vee_{i=1}^{n} c_i$, where $c_i = \wedge_{j=1}^{i_m} e_{ij}$, if

$$\text{LLM}(q) = \bigvee_{i=1}^{n} \left( \bigwedge_{j=1}^{i_m} \text{LLM}(e_{ij}) \right).$$

Here, $e_{ij}$ is an atomic relation fact for any $1 \leqslant i \leqslant n$ and $1 \leqslant j \leqslant i_m$.

*Proof.* Applying the definition of consistency (Definition 2) on the LLM response to $q$, we obtain

$$\text{LLM}(q) = \text{LLM}(c_1) \vee \text{LLM}(c_2) \vee \ldots \vee \text{LLM}(c_n)$$

$$= \text{LLM}(\bigwedge_{j=1}^{1_m} e_{1j}) \vee \text{LLM}(\bigwedge_{j=1}^{2_m} e_{2j}) \vee \ldots \vee \text{LLM}(\bigwedge_{j=1}^{n_m} e_{nj})$$

$$= \left( \bigwedge_{j=1}^{1_m} \text{LLM}(e_{1j}) \right) \vee \left( \bigwedge_{j=1}^{2_m} \text{LLM}(e_{2j}) \right) \vee \ldots \vee \left( \bigwedge_{j=1}^{n_m} \text{LLM}(e_{nj}) \right)$$

$$= \bigvee_{i=1}^{n} \left( \bigwedge_{j=1}^{i_m} \text{LLM}(e_{ij}) \right)$$

$\square$

**Proposition 2.** (I) An LLM is consistent on a simple atomic fact if it is accurate both on the fact and its negation. (II) For a complex DNF fact, the LLM is consistent if it is accurate on the DNF fact as well as on all constituent atomic facts.

*Proof.* (I) Given a fact $q$ and its truth value $\mathbf{T}(q) \in \{0, 1\}$, an LLM response $\text{LLM}(q)$ is correct if $\text{LLM}(q) = \mathbf{T}(q)$ (Definition 1). Similarly, for the negation fact $\neg q$, an LLM response $\text{LLM}(\neg q)$ is correct if $\text{LLM}(\neg q) = \mathbf{T}(\neg q)$. Since $\neg q$ is the negation of $q$,

$$\mathbf{T}(q) = \neg \mathbf{T}(\neg q)$$
$$\implies \text{LLM}(q) = \neg \text{LLM}(\neg q)$$
$$\implies \neg \text{LLM}(q) = \text{LLM}(\neg q)$$
$$\implies \text{LLM is consistent on } (q, \neg q)$$

The last line follows from the definition of consistency (Definition 2)

(II) For a DNF fact $q = \bigvee_{i=1}^{n} \left( \bigwedge_{j=1}^{i_m} e_{ij} \right)$, we similarly assume that the LLM accurately classifies $q$ and its constituent atomic fact $e_{ij}$. That is, $\text{LLM}(q) = \mathbf{T}(q)$ and $\text{LLM}(e_{ij}) = \mathbf{T}(e_{ij})$. Relating $q$ with its constituents $e_{ij}$ as $q = \bigvee_{i=1}^{n} \left( \bigwedge_{j=1}^{i_m} e_{ij} \right)$, we derive the following.

$$\mathbf{T}(q) = \bigvee_{i=1}^{n} \left( \bigwedge_{j=1}^{i_m} \mathbf{T}(e_{ij}) \right)$$

$$\implies \text{LLM}(q) = \bigvee_{i=1}^{n} \left( \bigwedge_{j=1}^{i_m} \text{LLM}(e_{ij}) \right).$$

Therefore, the LLM is consistent to the DNF fact according to Proposition 1.

$\square$

**Proposition 3.** An LLM is consistent with respect to a CNF fact $q = \wedge_{i=1}^{n} c_i$, where $c_i = \vee_{j=1}^{i_m} e_{ij}$, if

$$\text{LLM}(q) = \bigwedge_{i=1}^{n} \left( \bigvee_{j=1}^{i_m} \text{LLM}(e_{ij}) \right).$$

Here, $e_{ij}$ is an atomic relation fact for any $1 \leqslant i \leqslant n$ and $1 \leqslant j \leqslant i_m$.

*Proof.* Applying the definition of consistency to the LLM response to $q$, we obtain

$$\text{LLM}(q) = \text{LLM}(c_1) \wedge \text{LLM}(c_2) \wedge \ldots \wedge \text{LLM}(c_n)$$

$$= \text{LLM}(\bigvee_{j=1}^{1_m} e_{1j}) \wedge \text{LLM}(\bigvee_{j=1}^{2_m} e_{2j}) \wedge \ldots \wedge \text{LLM}(\bigvee_{j=1}^{n_m} e_{nj})$$

$$= \left(\bigvee_{j=1}^{1_m} \text{LLM}(e_{1j})\right) \wedge \left(\bigvee_{j=1}^{2_m} \text{LLM}(e_{2j})\right) \wedge \ldots \wedge \left(\bigvee_{j=1}^{n_m} \text{LLM}(e_{nj})\right)$$

$$= \bigwedge_{i=1}^{n} \left(\bigvee_{j=1}^{i_m} \text{LLM}(e_{ij})\right)$$

$\square$

## D  ADDITIONAL DISCUSSIONS ON METHODOLOGY

---
**Algorithm 2** Supervised fine-tuning for logical consistency

---
    **Input:** A language model LLM, facts $\mathcal{R}$
    **Output**: fine-tuned model LLM*
1: $\mathcal{D}_{\texttt{instruct}} \leftarrow \texttt{BuildDataset}(\mathcal{R})$      ▷ Create an instruction dataset from knowledge queries
2: LLM* $\leftarrow \texttt{SupervisedFinetuning}(\text{LLM}, \mathcal{D}_{\texttt{instruct}})$

---

### D.1  SUPERVISED FINE-TUNING

We adopted a parameter-efficient fine-tuning (PEFT) method based on QLoRA (Quantized Low-Rank Adaptation) (Dettmers et al., 2024), which is a more memory-efficient adaptation of LoRA (Hu et al., 2021a). The main idea behind LoRA is to freeze the pre-trained model weight-matrix $W_0$ of size $d \times d$ and represent the trainable update-matrix ($\Delta W$) as two low-rank matrices $A$ and $B$ of dimensions $d \times r$ and $r \times d$, respectively. Here, $r << d$, e.g., $r = 64$ and $d \approx 16\text{K}$ for Llama2-7B and $d \approx 8\text{K}$ for Gemma-2B in our experiments. Given an input vector $x$, during forward-pass, instead of computing $(W_0 + \Delta W)x$, LoRA computes $(W_0 x + BAx)$. The benefit of storing two low-rank matrices instead of $\Delta W$ is that the model needs to update only $2rd$ parameters, which is more efficient than updating $d^2$ parameters. QLoRA quantizes the precision of the weights to 4 bits. Due to quantization, QLoRA is more memory efficient; for instance, it has been reported that one can fine-tune a 65B parameter Llama model on a single 48GB GPU while preserving full 16-bit finetuning task performance (Dettmers et al., 2024).

## E  DATASETS

### E.1  DATA SOURCES

We consider existing KGs, such as Freebase, NELL, and WikiKG90Mv2, and convert to logical fact-checking datasets. Freebase contains real-world entities such as people, places, institutions, and their relations from the Freebase KG. NELL is a dataset built from the Web via an intelligent agent called Never-Ending Language Learner. This agent attempts to learn over time to read the web. NELL has accumulated over 100 million candidate triplets by reading the web (April 2018) with different levels of confidence. WikiKG90Mv2 is a KG extracted from the entire Wikidata knowledge base.

### E.2  OUR CURATED DATASETS

Our curated datasets contain the following logical facts: simple facts $p$ and $\neg p$, facts with one operators $p \wedge q$ and $p \vee q$, complex facts with two operators $p \wedge (q \vee r)$ and $p \vee (q \wedge r)$, and commutative rules $p \vee q \leftrightarrow q \vee p$ and $p \wedge q \leftrightarrow q \wedge p$. In fine-tuning, we split each fact type into training, evaluation, and test sets having 1K, 5K, and 5K samples, respectively (Table 5). For

Table 4: Statistics of KGs (left) and simple facts obtained from them (right)

| KG | #Entities | #Relations | #Triplets (simple facts) |
|---|---|---|---|
| FB15K | $14,951$ | $1,345$ | $592,213$ |
| NELL | $63,361$ | $200$ | $142,804$ |
| Wiki | $9,12,30,610$ | $1,387$ | $60,10,62,811$ |

---

**LLMQuery**

$\langle\langle$SYS$\rangle\rangle$ You are a helpful assistant. Always follow the instructions precisely and output the response exactly in the requested format. $\langle\langle$/SYS$\rangle\rangle$

Consider the context as a set of triplets where entries are separated by '|' symbol. Answer question according to the context.

T.S. Eliot | award winner | Nobel Prize in Literature
Thomas Mann | influenced | Franz Kafka
...
Thomas Mann | award winner | Nobel prize in Literature
...

Do not add additional text. Is the following triplet factually correct? Answer with Yes or No.

Thomas Mann | award winner | Nobel prize in Literature

{Yes or No}

---

Figure 3: (Continuing Figure 1) A representative `LLMQuery` on a simple fact used in our experiments. The expected LLM response is in blue color.

instance, FreebaseLFC contains 1K facts from each of the 4 fact-types: $p$, $\neg p$, $p \wedge q$, $p \vee q$ resulting in $1K \times 4 = 4K$ facts. In fine-tuning, we only consider training facts from FreebaseLFC, hence the other datasets do not have any additional training facts. In Wiki dataset, we randomly sample 10 million triplets to prepare subgraph creation via BFS and training/evaluation/test split.

### E.3 MORE COMPLEX AND HIGHER-ORDER LOGIC DATASETS

We have curated two more datasets from Freebase containing more complex logical expressions, especially the law of Syllogism (LoS) and the first-order logic (FoL). In the LoS dataset, there are 100 queries of the form $r_1(p, q) \wedge r_2(q, s)$ encapsulating the logic $(p \Rightarrow q) \wedge (q \Rightarrow s)$, which, according to the law of Syllogism should be equivalent to $p \Rightarrow s$.

---

Consider the context: $Context.

Is the fact correct given the context? $Fact-1 and $Fact-2.

LLMQuery on ($Fact-1 $\wedge$ $Fact-2, $Context)

---

Consider the context: $Context.

Is the fact correct given the context? $Fact-1 or $Fact-2.

LLMQuery on ($Fact-1 $\vee$ $Fact-2, $Context)

---

Figure 4: (Continuing Figure 1) `LLMQuery` when considering conjunctive and disjunctive facts. We replace logical operators with their natural language description, such as $\wedge$ with 'and' and $\vee$ with 'or', when constructing `LLMQuery`.

Table 5: Fact-checking datasets

| Datasets | Fact types and Rules | Training | Evaluation | Test |
|---|---|---|---|---|
| FreebaseLFC | $p, \neg p, p \wedge q, p \vee q$ | 1K ($\times 4$) | 5K ($\times 4$) | 5K ($\times 4$) |
| | $p \wedge (q \vee r), p \vee (q \wedge r), p \vee q \leftrightarrow q \vee p, p \wedge q \leftrightarrow q \wedge p$ | — | 5K ($\times 4$) | 5K ($\times 4$) |
| NELLLFC | All above fact types and rules | — | 5K ($\times 8$) | 5K ($\times 8$) |
| WikiLFC | $p, \neg p$ | — | 5K ($\times 2$) | 5K ($\times 2$) |

**Example:** `Contains`(Oceania, New Zealand) $\wedge$ `Country`(New Zealand, Auckland) is logically equivalent to `Contains`(Oceania, Auckland).

In the FoL dataset, there are 500 queries of the form $\exists_x r_1(p, x) \wedge r_2(x, q)$ and their negations in the form of $\forall_x \neg r_1(p, x) \vee \neg r_2(x, q)$. The negation of the former should be logically equivalent to the latter. In `LLMQuery`, we consider the natural language description 'There exists an entity $x$ such that' and 'For all entity $x$ such that' to express existential $\exists$ and for-all $\forall$ quantifiers, respectively.

## F    PARAMETER SETTINGS AND SYSTEM CONFIGURATIONS

### F.1    PARAMETER SETTINGS

For an efficient fine-tuning, we adopted QLoRA implementation from Huggingface (Dettmers et al., 2024). In Llama2-7B and 13B, the hyperparameter choices of QLoRA are the following: learning rate $2 \times 10^{-5}$, weight decay 0.001, warm-up ratio 0.03, batch size 8, $r = 16$, $\alpha = 16$, and dropout 0.1. In Gemma-2B, we consider learning rate $2 \times 10^{-6}$ and keep other hyper-parameters similar to Llama. For high-throughput and memory-efficient inference, we adopt the vLLM library (Kwon et al., 2023), and set temperature to 0 for a deterministic output. We limit the context length in the LLMQuery by selecting relevant triplets of around 1000 tokens.

### F.2    SYSTEM SETUP

All experiments are conducted in Python version 3.8.0. Fine-tuning is conducted on a cluster with two NVIDIA A40 45 GB GPUs having Intel(R) Xeon(R) Gold 5317 CPU @ 3.00 GHz, 48 core and 1007 GB RAM. Inference is conducted on a cluster with two Tesla V100-PCIE 32 GB GPUs having Intel(R) Xeon(R) Gold 6134M CPU @ 3.20 GHz, 32 core, and 755 GB RAM.

## G    ADDITIONAL EXPERIMENTS

### G.1    EFFICIENCY

We compare the per-epoch running time of PEFT vs. full fine-tuning of Llama2-7B (Table 6). PEFT requires on average 4.43 hours to complete an epoch including the computation of evaluation loss, whereas full fine-tuning requires 12.52 hours. *Therefore, PEFT is more efficient than full fine-tuning in improving the efficiency of supervised fine-tuning for logical consistency.*

Table 6: Running time of PEFT vs. Full Fine-tuning of Llama2-7B

| | PEFT | Full fine-tuning |
|---|---|---|
| Time per epoch | 4.43 hours | 12.52 hours |

Table 7 shows the end-to-end inference time for facts with different number of operators. Increasing the number of operators results in a higher time for BFS and subsequent context creation time. In particular, the BFS time for FreebaseLFC is higher than that in NELLLFC, and WikiLFC has the lowest BFS time. BFS time is correlated with the sparsity of the KG and can be approximated by how many edges are extracted in the BFS subgraph – for simple facts, the median number of edges

Table 7: Per-sample end-to-end fact-checking time in seconds using Llama2-7B

| Dataset | Fact | BFS | Context Creation | LLM Inference | Total |
|---------|------|-----|------------------|---------------|-------|
| FreebaseLFC | $p, \neg p$ | $0.04 \pm 0.10$ | $0.06 \pm 0.10$ | $0.21 \pm 0.00$ | $0.30 \pm 0.14$ |
| | $p \wedge q$ | $0.07 \pm 0.15$ | $0.15 \pm 0.18$ | $0.22 \pm 0.00$ | $0.44 \pm 0.25$ |
| | $p \vee (q \wedge r)$ | $0.09 \pm 0.20$ | $0.32 \pm 0.34$ | $0.22 \pm 0.01$ | $0.63 \pm 0.39$ |
| NELLLFC | $p, \neg p$ | $0.02 \pm 0.08$ | $0.04 \pm 0.05$ | $0.19 \pm 0.00$ | $0.24 \pm 0.10$ |
| | $p \wedge q$ | $0.04 \pm 0.07$ | $0.08 \pm 0.13$ | $0.21 \pm 0.00$ | $0.32 \pm 0.16$ |
| | $p \vee (q \wedge r)$ | $0.05 \pm 0.15$ | $0.09 \pm 0.09$ | $0.22 \pm 0.00$ | $0.36 \pm 0.18$ |
| WikiLFC | $p, \neg p$ | $0.01 \pm 0.05$ | $0.01 \pm 0.01$ | $0.08 \pm 0.00$ | $0.09 \pm 0.06$ |

in the BFS subgraph is 9773, 763.5, and 7 for FreebaseLFC, NELLLFC, and WikiLFC, respectively. Further, since WikiLFC is the sparsest among all KGs in our experiments, the context creation time and subsequent LLM inference time is lower in WikiLFC compared to other two datasets – for example, the LLM inference time is the lowest in WikiLFC (0.08 seconds) compared to FreebaseLFC and NELLLFC ($\sim 0.20$ seconds). *In summary, the end-to-end fact-checking time is efficient (less than* 0.63 *seconds per sample) in our implementation*.

### G.2 IMPACT OF KG CONTEXTS

Table 8: Impact of using context in the LLMQuery on the accuracy and logical consistency of simple facts in FreebaseLFC dataset. Results are for the base model without fine-tuning.

| Model | Dataset | Accuracy | | Logical Consistency | |
|-------|---------|----------|--|---------------------|--|
| | | Without Context | With Context | Without Context | With Context |
| Llama2-7B | FreebaseLFC | 0.51 | **0.88** | 0.03 | **0.77** |

Table 8 shows the effectiveness of using a context on the accuracy and logical consistency of Llama2-7B for simple facts. Adding a context provides the LLM a premise to answer the subsequent fact. As such, Llama2-7B base model demonstrates a higher accuracy and logical consistency in the presence of a context, and *hence, in all our experiments we have provided a KG context to the LLMQuery*.

### G.3 IMPACT OF CONTEXT LENGTH

The inference performance of LLMs is affected by the length of the context of the underlying prompt. In our study, we vary the context length of the KG context using 1000 (default) and 200 tokens and report the performance of logical consistency and accuracy before and after fine-tuning in the FreebaseLFC dataset (Table 9). Limited context results in an improved accuracy on the base model (before fine-tuning). However, the consistency is in-general unaffected due to reduced context length. Upon fine-tuning the model as proposed in the paper, both accuracy and consistency improve with reduced context length. *Therefore, context length provides a useful knob to improve the accuracy of an LLM in answering fact-checking queries, whereas to improve logical consistency, supervised fine-tuning is still needed.*

### G.4 IMPACT OF DIFFERENT CONTEXT RETRIEVAL METHODS

In addition to the BFS-based context retrieval method in §3.4, we consider two alternative retrieval methods to compare their effectiveness in providing the best context to the LLM and the resultant consistency of the LLM in answering fact-checking queries.

- **BFS+Vector embedding method.** We compute 2-hop neighbours by BFS to extract relevant triplets to the query. The relevant triplets are first embedded using the Sentence Transformer (Reimers & Gurevych, 2019). Finally, we prune the irrelevant triplets by applying nearest neighbour search on the embeddings and selecting the top-$k$ relevant triplets

Table 9: Impact of reduced context length in FreebaseLFC dataset.

| Model | Context length | Fact | Accuracy | | Logical Consistency | |
|---|---|---|---|---|---|---|
| | | | **Before FT** | **After FT** | **Before FT** | **After FT** |
| Gemma-2B | 1000 tokens | $p, \neg p$ | 0.82 | **0.98** | 0.66 | **0.96** |
| | | $p \wedge q$ | 0.45 | **0.83** | 0.70 | **0.84** |
| | | $p \vee q$ | 0.73 | **0.94** | 0.91 | 0.90 |
| | 200 tokens | $p, \neg p$ | 0.92 | **1.00** | 0.84 | **0.99** |
| | | $p \wedge q$ | 0.50 | **0.98** | 0.63 | **0.97** |
| | | $p \vee q$ | 0.75 | **0.93** | 0.90 | **0.93** |

with the highest relevance scores (w.r.t the query) to construct the KG Context. Here, relevance scores are computed based on the $L_2$ norm of the embedding vector for the query subtracted by the embedding vector for the triplet.

- **Vector Embedding Method.** We bypass the BFS step, rather compute and index the embeddings of all the triplets in the KG, such as Freebase. Finally, we extract relevant triplets by applying nearest neighbour search on the embeddings and selecting top-$k$ triplets with the highest relevance scores (w.r.t. the query) to construct the KG Context.

Table 10: Impact different retrieval methods. Experiments are performed on simple facts from the FreebaseLFC dataset and on Llama2-7B model.

| Retrieval Method | Accuracy | | Logical Consistency | |
|---|---|---|---|---|
| | **Before FT** | **After FT** | **Before FT** | **After FT** |
| BFS + Relation hierarchy (§3.4) | 0.88 | **0.97** | 0.76 | **0.94** |
| BFS + Vector Embedding | 0.92 | **0.96** | 0.85 | **0.92** |
| Vector Embedding (without BFS) | 0.89 | **0.98** | 0.78 | **0.96** |

We observe that before fine-tuning, accuracy and consistency of BFS + embedding based retrieval is better than that of retrieving context from relation hierarchy and ontology applied to BFS subgraph of 2-hop neighbours. In addition, bypassing BFS and only applying vector embedding results in a similar performance as BFS + relation hierarchy. Thus, vector embedding appears as an effective way to provide context to the LLM.

We further verify that the supervised fine-tuning can improve both accuracy and consistency in all retrieval methods. Note that we reuse the fine-tuned model on context from BFS + relation hierarchy and apply it on data from the vector-embedding based retrieval methods. In preprocessing, embedding based retrieval has a higher preprocessing time (almost 100x than the retrieval time based on relation hierarchy) to embed the subgraph than relation hierarchy based retrieval. However, such vector embeddings can be performed offline.

**Handling Dynamically Varying Contexts.** Real-world applications may need context retrieval to quickly adapt to new information, e.g., the knowledge graph evolving with time. In the case of evolving KGs, the "BFS + Relation hierarchy" retrieval is a computationally better alternative than embedding-based retrieval methods such as "BFS + Vector Embedding" or "Vector Embedding (without BFS)" method. Although the embedding-based methods slightly improve the accuracy and consistency (as shown in Table 10), due to their higher pre-processing overhead, they must be conducted offline repeatedly as the KG changes to obtain better entity embeddings.

## G.5 Results with Zero-Shot, 2-shot and Chain-of-thought Prompting

### G.5.1 Supervised Fine-tuning vs Zero-shot Prompting

We demonstrate the extended results on comparing instruction prompting with fine-tuning in Table 11. We find that fine-tuning is superior to instruction prompting on simple facts, across datasets and models.

Table 11: Impact of instruction prompting (Prompt) vs. fine-tuning (FT) on simple facts. Fine-tuning results in higher accuracy and consistency than instruction prompting.

| Model | Dataset | Accuracy | | | Logical Consistency | | |
|---|---|---|---|---|---|---|---|
| | | Before FT | Prompt | After FT | Before FT | Prompt | After FT |
| Llama2-13B | FreebaseLFC | 0.90 | 0.89 | **0.93** | 0.81 | 0.79 | **0.86** |
| | NELLLFC | 0.88 | 0.85 | **0.97** | 0.76 | 0.72 | **0.93** |
| | WikiLFC | 0.96 | 0.94 | **0.96** | 0.92 | 0.90 | **0.93** |
| Llama2-7B | FreebaseLFC | 0.87 | 0.82 | **0.97** | 0.76 | 0.64 | **0.94** |
| | NELLLFC | 0.71 | 0.61 | **0.94** | 0.44 | 0.23 | **0.88** |
| | WikiLFC | 0.90 | 0.82 | **0.90** | 0.80 | 0.64 | **0.81** |
| Gemma-2B | FreebaseLFC | 0.82 | 0.88 | **0.98** | 0.66 | 0.77 | **0.96** |
| | NELLLFC | 0.81 | 0.81 | **0.94** | 0.62 | 0.63 | **0.89** |
| | WikiLFC | 0.76 | 0.69 | **0.90** | 0.52 | 0.43 | **0.80** |

### G.5.2 2-shot prompting

Table 12: Zero-shot, 2-shot and Chain-of-Thought (CoT) prompting results on Llama2-7B model.

| Dataset | Accuracy | | | | Logical Consistency | | | | |
|---|---|---|---|---|---|---|---|---|---|
| | Before FT | 0-shot | 2-shot | CoT | After FT | Before FT | 0-shot | 2-shot | CoT | After FT |
| FreebaseLFC | 0.87 | 0.89 | 0.78 | 0.93 | **0.97** | 0.76 | 0.79 | 0.58 | 0.86 | **0.94** |

In Table 3, we showed that for our task, supervised fine-tuning produces better consistency and accuracy compared to zero-shot prompts. We have also tested 2-shot prompts to understand whether providing examples (one positive and one negative example) to the LLM can improve its performance on fact-checking consistency (see Figure 5). The prompt is shown in the following text box along with the accompanying result in Table 12. In the 2-shot prompt, we denote the example contexts in red color and the example facts in blue color.

We observe that 2-shot prompting performs worse than zero-shot prompting. This is due to the reason we mentioned in §4.1: There are nuances and intricacies in logical facts, and providing examples only adds to the complexity rather than helping the LLM understand the differences between true and false facts. Such results are not unprecedented in the literature. For instance, Zhao et al. (2021) showed that there are biases in GPT-3 and GPT-2, e.g., their tendency to output recent or common tokens which caused these models to produce drastically different accuracy on various tasks, depending on the prompt format and ordering of examples in the prompt. It was also shown that in some of the text-classification task datasets, 2-shot prompting had worse performance than zero-shot prompting (see Table 1 in Zhao et al. (2021)).

### G.5.3 Chain-of-Thought (CoT) Prompting.

We have created a chain-of-thought (CoT) prompt to test whether providing CoT to the LLM can improve its performance on fact-checking consistency (see Figure 6). Similar to the 2-shot prompt, the CoT prompt contains a positive example and a negative example. A positive example is one where the context contains the triplets in the logic query, and a negative example is where the logic query is absent from the context. In addition to the examples, we also provide a reasoning behind the correct answer. Table 12 presents the result for CoT prompting.

---

**2-shot Prompt**

⟨⟨SYS⟩⟩ You are a helpful assistant. Always follow the instructions precisely and output the response exactly in the requested format. ⟨⟨/SYS⟩⟩

# Example 1:

Consider the context as a set of triplets where entries are separated by '|'. Answer the question according to the context.

Context:
County Durham | contains reverse | England
Hartlepool | contains reverse | County Durham
Hartlepool | administrative parent | County Durham
...

Question:
Hartlepool | contains reverse | County Durham

Answer: Yes

# Example 2:

Consider the context as a set of triplets where entries are separated by '|'. Answer the question according to the context.

Context:
County Durham | contains reverse | England
Hartlepool | contains reverse | County Durham
Hartlepool | administrative parent | County Durham
...

Question:
Hartlepool | contains reverse | Cottam, East Riding of Yorkshire

Answer: No

---

**Test Context & Fact**

Consider the context as a set of triplets where entries are separated by '|'. Answer the question according to the context.

Context: $Context
Question: $Fact
Answer: {Yes or No}

---

Figure 5: Example of 2-shot prompt, where representative positive and negative examples are provided to guide the LLM to answer the target fact.

As we compare the results of CoT with "After FT", we conclude that CoT does not demonstrate any improvement over fine-tuning in our specific case. We hypothesize that achieving logical consistency requires an update in the internal weights of the LLM, such as via supervised fine-tuning, and simply providing examples is insufficient.

### G.6 IMPACT OF LEARNING RATE

While fine-tuning Llama2-7B, we consider different learning rates (Figure 7). A higher learning rate such as $2 \times 10^{-4}$ quickly decreases training loss and results in an over-fitting – as such, there is an increase in evaluation loss over epochs. On the other hand, a lower learning rate $2 \times 10^{-7}$ neither decreases training loss nor evaluation loss. Finally, both learning rates $2 \times 10^{-5}$ and $2 \times 10^{-6}$ balance between train/evaluation loss; and we select $2 \times 10^{-5}$ as the final learning rate for subsequent

---

**Chain of Thought Prompt**

⟨⟨SYS⟩⟩ You are a helpful assistant. Always follow the instructions precisely and output the response exactly in the requested format. ⟨⟨/SYS⟩⟩

Consider the context as a set of triplets where entries are separated by '|' symbol. Answer the question according to the context.

Hartlepool |administrative children reverse | County Durham
County Durham | contains reverse | England
Hartlepool | contains reverse | County Durham
County Durham | containedby reverse | Durham University
...

Do not add additional text. Is the following triplet factually correct? Answer with Yes or No.

Hartlepool | contains reverse | County Durham

The provided triplet 'Hartlepool | contains reverse | County Durham' appears in the context on the ninth line. The answer is Yes.

Consider the context as a set of triplets where entries are separated by '|' symbol. Answer the question according to the context.

Hartlepool | administrative children reverse | County Durham
County Durham | contains reverse | England
Hartlepool | contains reverse | County Durham
County Durham | containedby reverse | Durham University
...

Do not add additional text. Is the following triplet factually correct? Answer with Yes or No.

Hartlepool | contains reverse | Cottam, East Riding of Yorkshire

The provided triplet 'Hartlepool | contains reverse | Cottam, East Riding of Yorkshire' does not appear in any line of the context. The answer is No.

**Test Context & Fact**

Consider the context as a set of triplets where entries are separated by '|' symbol. Answer the question according to the context.

$Context

Do not add additional text. Is the following triplet factually correct? Answer with Yes or No.

$Fact

{Reasoning followed by Yes or No answer}

Figure 6: Example of chain of thought (CoT) prompt, where representative positive and negative examples are followed by CoT reasoning and answers.

experiments in Llama2-7B due to achieving lower evaluation loss. *Therefore, learning rate provides precise control over generalization performance in fine-tuning.*

## G.7 ACCURACY AND LOGICAL CONSISTENCY ACROSS EPOCHS IN FINE-TUNING.

We demonstrate how accuracy and logical consistency vary across fine-tuning epochs in Llama2-13B (Figure 8), Llama2-7B (Figure 9), and Gemma-2B (Figure 10). We present results on FreebaseLFC dataset, which are used in training, and on NELLLFC dataset to understand the generalization in fine-tuning. In general, both accuracy and consistency increase as we increase the epochs across

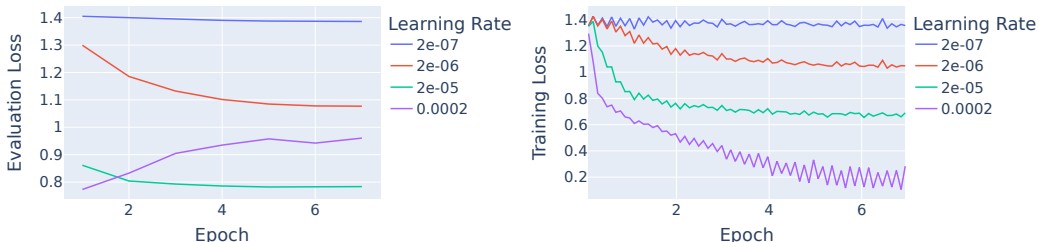

Figure 7: Impact of different learning rates in fine-tuning Llama2-7B model. Higher learning rate results in over-fitting and a degraded performance on the evaluation data.

different fact types: simple fact $p$, $\neg p$, and complex facts $p \wedge q$ and $p \vee q$. In most cases, accuracy and consistency surpass their initial values on the base models (shown in dashed horizontal lines). Further, accuracy and consistency are often correlated across epochs. Exceptionally in LLama2 models, accuracy and consistency occasionally decreases for $p \vee q$ fact, whereas Gemma-2B does not exhibit this phenomenon. *In summary, both accuracy and logical consistency improve over epochs in fine-tuning across models and datasets*.

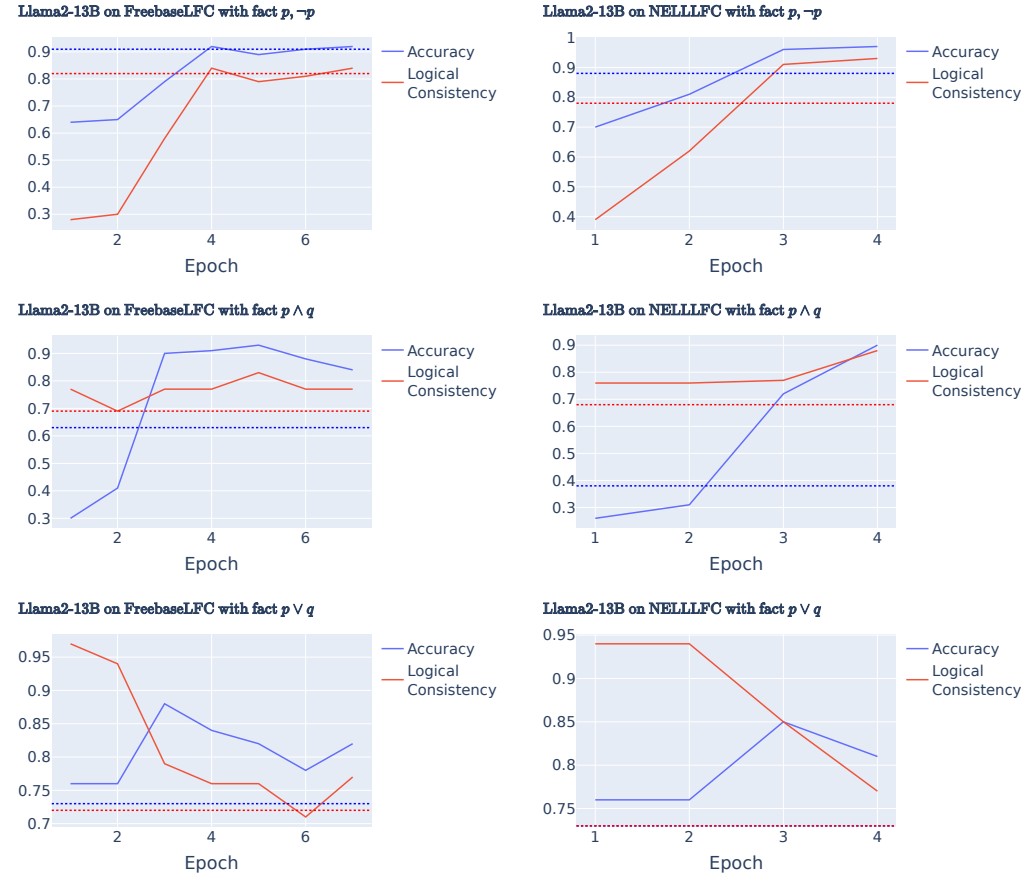

Figure 8: Accuracy and logical consistency of evaluation data across fine-tuning epochs in Llama2-13B. The left row denotes results on FreebaseLFC (included in training), while the right row denotes results on NELLLFC. Dashed line denotes accuracy and consistency before fine-tuning.

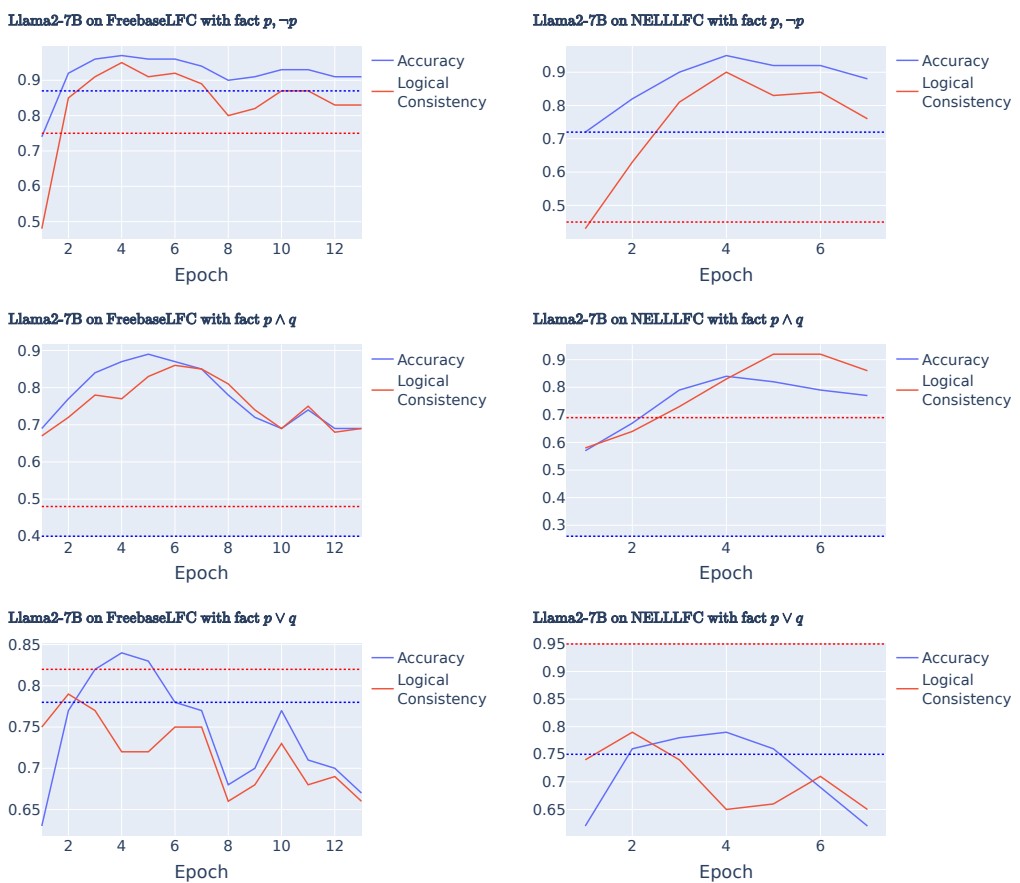

Figure 9: Accuracy and logical consistency of evaluation data across fine-tuning epochs in Llama2-7B. The left row denotes results on FreebaseLFC (included in training), while the right row denotes results on NELLLFC. Dashed line denotes accuracy and consistency before fine-tuning.

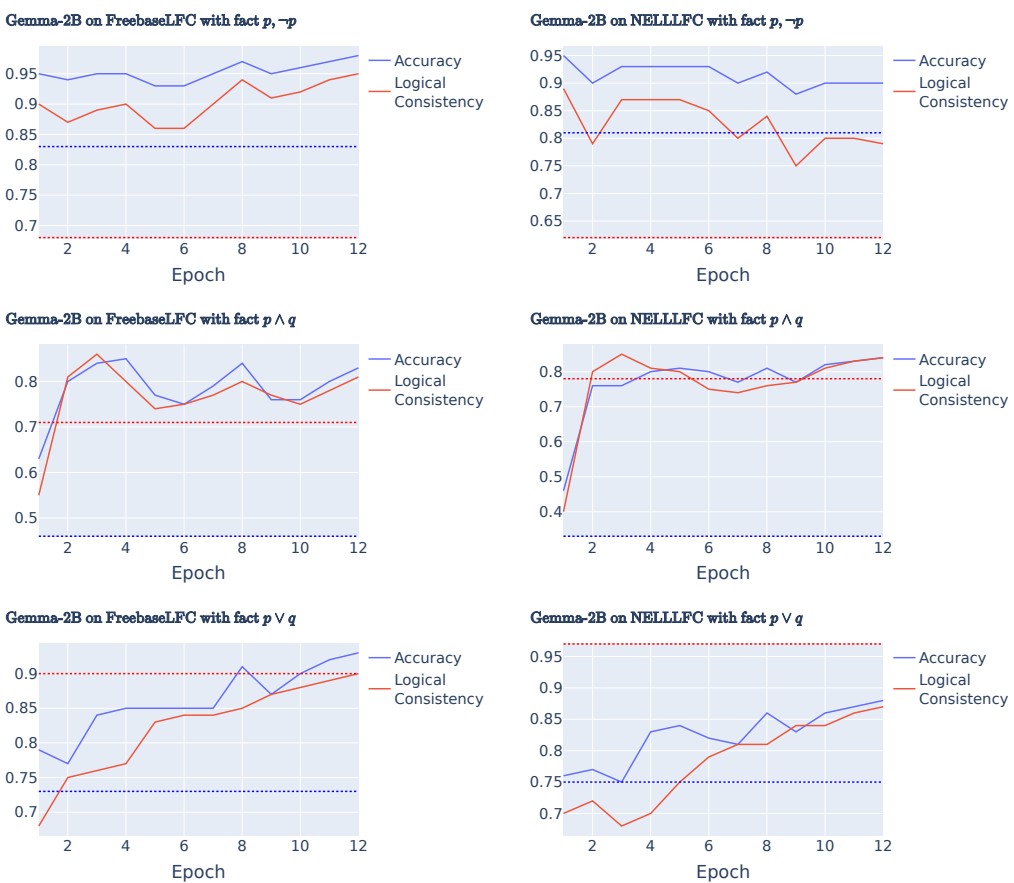

Figure 10: Accuracy and logical consistency of evaluation data across fine-tuning epochs in Gemma-2B. The left row denotes results on FreebaseLFC (included in training), while the right row denotes results on NELLLFC. Dashed line denotes accuracy and consistency before fine-tuning.

# H GENERALIZABILITY TO COMPLEX FACTS AND RULES

Table 13: Generalization of fine-tuning to complex facts and rules (Extended Table 2).

| Model | Dataset | Fact | Accuracy | | Logical Consistency | |
|---|---|---|---|---|---|---|
| | | | Before FT | After FT | Before FT | After FT |
| Llama2-13B | FreebaseLFC | $p \vee (q \wedge r)$ | 0.74 | **0.85** | 0.78 | **0.81** |
| | | $p \wedge (q \vee r)$ | 0.53 | **0.82** | 0.63 | **0.79** |
| | | $p \vee q \leftrightarrow q \vee p$ | 0.73 | **0.76** | 0.85 | **0.98** |
| | | $p \wedge q \leftrightarrow q \wedge p$ | 0.61 | **0.93** | 0.84 | **0.91** |
| | NELLLFC | $p \vee (q \wedge r)$ | 0.67 | **0.77** | 0.72 | **0.80** |
| | | $p \wedge (q \vee r)$ | 0.42 | 0.38 | 0.75 | **0.85** |
| | | $p \vee q \leftrightarrow q \vee p$ | 0.73 | **0.76** | 0.80 | **0.98** |
| | | $p \wedge q \leftrightarrow q \wedge p$ | 0.38 | **0.70** | 0.90 | 0.85 |
| Llama2-7B | FreebaseLFC | $p \vee (q \wedge r)$ | 0.68 | **0.74** | 0.78 | 0.71 |
| | | $p \wedge (q \vee r)$ | 0.42 | **0.74** | 0.54 | **0.75** |
| | | $p \vee q \leftrightarrow q \vee p$ | 0.78 | **0.84** | 0.87 | 0.85 |
| | | $p \wedge q \leftrightarrow q \wedge p$ | 0.39 | **0.89** | 0.91 | **0.92** |
| | NELLLFC | $p \vee (q \wedge r)$ | 0.63 | **0.70** | 0.94 | 0.76 |
| | | $p \wedge (q \vee r)$ | 0.37 | **0.66** | 0.80 | 0.79 |
| | | $p \vee q \leftrightarrow q \vee p$ | 0.75 | **0.78** | 0.98 | 0.82 |
| | | $p \wedge q \leftrightarrow q \wedge p$ | 0.26 | **0.81** | 0.99 | 0.95 |
| Gemma-2B | FreebaseLFC | $p \vee (q \wedge r)$ | 0.62 | **0.83** | 0.91 | 0.77 |
| | | $p \wedge (q \vee r)$ | 0.38 | **0.74** | 0.77 | **0.81** |
| | | $p \vee q \leftrightarrow q \vee p$ | 0.73 | **0.94** | 0.90 | **0.93** |
| | | $p \wedge q \leftrightarrow q \wedge p$ | 0.45 | **0.83** | 0.80 | **0.90** |
| | NELLLFC | $p \vee (q \wedge r)$ | 0.62 | **0.81** | 0.97 | 0.80 |
| | | $p \wedge (q \vee r)$ | 0.37 | **0.66** | 0.82 | 0.80 |
| | | $p \vee q \leftrightarrow q \vee p$ | 0.75 | **0.88** | 0.99 | 0.93 |
| | | $p \wedge q \leftrightarrow q \wedge p$ | 0.33 | **0.83** | 0.92 | 0.90 |

## H.1 DE-MORGAN'S LAWS

We have tested the generalizability of the fine-tuned models on more challenging logic rules such as De-Morgan's laws. In order to finetune Llama2-13B, we have incorporated in the training datasets simple facts, conjunction, disjunction, first-order logic rules, law of syllogism, and a small number of De Morgan's law rules. The results are shown in Table 14.

We observe that before any fine-tuning, the accuracy of De-Morgan's law on conjunctive ($\neg(p \wedge q) \leftrightarrow \neg p \vee \neg q$) and disjunctive ($\neg(p \vee q) \leftrightarrow \neg p \wedge \neg q$) facts is the same as a random guess model while being highly consistent. This is because the base Llama2-13B model classifies each fact (both sides of $\leftrightarrow$) as true, resulting in a consistent yet inaccurate fact-checker. Upon fine-tuning, the accuracy on both conjunctive and disjunctive fact-types improves significantly, while retaining a similar performance in logical consistency. Therefore, our experiments demonstrate the generality of applying supervised fine-tuning in improving both logical consistency and accuracy in logical fact-checking across multiple fact types, including De Morgan's law.

## H.2 FIRST-ORDER LOGIC AND LAW OF SYLLOGISM

We demonstrate results on first-order logic (FoL) and law of syllogism (LoS) queries when considering only proposition logic queries for fine-tuning in Table 15 and all propositional logic, FoL, and LoS queries for fine-tuning in Table 16. When all queries are considered, we observe higher accuracy and logical consistency on unseen FoL and LoS queries.

Table 14: Evaluating Generalizability of fine-tuned Llama2-13B using De-Morgan's law before and after fine-tuning. Experiments are conducted on FreebaseLFC dataset.

| | Accuracy | | Logical Consistency | |
|---|---|---|---|---|
| **Fact** | **Before FT** | **After FT** | **Before FT** | **After FT** |
| $\neg(p \vee q) \leftrightarrow \neg p \wedge \neg q$ | 0.25 | **0.99** | 1.00 | 0.98 |
| $\neg(p \wedge q) \leftrightarrow \neg p \vee \neg q$ | 0.75 | **0.97** | 0.99 | 0.95 |

Table 15: Performance on more complex logic settings: first-order logic (FoL) with an existential quantifier ($\exists$) and the law of syllogism (LoS), curated from the FreebaseLFC benchmark. FT results are on propositional logic benchmark.

| | | | Accuracy | | Logical Consistency | |
|---|---|---|---|---|---|---|
| **Model** | **Dataset** | **Fact** | **Before FT** | **After FT** | **Before FT** | **After FT** |
| Llama2-13B | FreebaseLFC | FoL | 0.57 | **0.66** | 0.13 | **0.37** |
| | | LoS | 0.68 | **0.95** | 0.73 | **0.91** |
| Llama2-7B | FreebaseLFC | FoL | 0.50 | **0.61** | 0.00 | **0.45** |
| | | LoS | 0.55 | **0.77** | 0.45 | **0.55** |
| Gemma-2B | FreebaseLFC | FoL | 0.50 | **0.57** | 0.00 | **0.80** |
| | | LoS | 0.55 | **0.91** | 0.76 | **0.82** |

## I PERFORMANCE ON NATURAL TEXT FACT-CHECKING DATASET

It is important to assess how well the proposed method translates to real-world situations where queries are of textual nature. In particular, does the supervised fine-tuning still generalize to textual data as opposed to triple context and query? We experiment with a widely known real-world fact-checking benchmark containing textual facts: FEVER (Fact Extraction and VERification) (Thorne et al., 2018).

FEVER consists of 185,445 claims generated by altering sentences extracted from Wikipedia, which are subsequently verified without knowledge of the sentence they were derived from. The claims are classified as *supported*, *refuted*, or *not-enough-info*. We consider supported and refuted claims as simple query $p$ and manually create natural language negation of each claim ($\neg p$) to assess for logical consistency. Our augmented FEVER dataset contains 580 supported claims and 220 refuted claims, totaling $580 + 220 = 800$ simple claims. We split these 800 simple claims into 400-200-200 for training, validation, and test, respectively, to perform supervised fine-tuning.

We adopt the Gemma-2B-it model (Team et al., 2024) as a proof of concept. In order to create the context, we retrieve 5 most relevant claims to the given query claim using vector embedding retrieval method. Table 17 shows that our method of improving logical consistency also works well in real-world situations where both queries and context to answer the queries are in natural text. In

Table 16: Performance on more complex logic settings: first-order logic (FoL) with an existential quantifier ($\exists$) and the law of syllogism (LoS), curated from the FreebaseLFC benchmark. FT results are on propositional logic, FoL, and LoS queries.

| | | | Accuracy | | Logical Consistency | |
|---|---|---|---|---|---|---|
| **Model** | **Dataset** | **Fact** | **Before FT** | **After FT** | **Before FT** | **After FT** |
| Llama2-13B | FreebaseLFC | FoL | 0.57 | **0.94** | 0.13 | **0.88** |
| | | LoS | 0.68 | **0.97** | 0.73 | **0.94** |
| Llama2-7B | FreebaseLFC | FoL | 0.50 | **0.91** | 0.00 | **0.86** |
| | | LoS | 0.55 | **0.98** | 0.45 | **0.97** |
| Gemma-2B | FreebaseLFC | FoL | 0.50 | **0.69** | 0.00 | **0.69** |
| | | LoS | 0.55 | **0.83** | 0.76 | 0.67 |

summary, the LLM (prior to fine-tuning) still lacks logical consistency and accuracy on natural text facts. Our supervised fine-tuning improves both the logical consistency and accuracy of the LLM and hence also generalizes to textual data.

Table 17: Improvement in accuracy and consistency on the FEVER dataset

| Model | Fact | Accuracy | | Logical Consistency | |
|---|---|---|---|---|---|
| | | Before FT | After FT | Before FT | After FT |
| Gemma-2B | $p, \neg p$ | 0.73 | **0.95** | 0.95 | **0.96** |

## J  EXPERIMENTAL RESULTS ON ADDITIONAL BASELINE: MINICHECK

We present the results by considering state-of-the-art fact-checking methods using KGs such as MiniCheck (Tang et al., 2024) in Table 18. The key technique employed in MiniCheck is to use GPT-4 and then construct synthetic training data having challenging instances and factual errors. Training on this data teaches the model, MiniCheck-FT5, to recognize complex information across the input claim and check each fact in it. When compared against our fine-tuned version of the Llama2-7B model over the FreebaseLFC dataset, we find that Llama2-7B shows better accuracy and consistency than MiniCheck-FT5. The reason behind MiniCheck's underwhelming performance is that it is designed for fact-checking on grounding documents, not for verifying propositional logic queries unlike ours, neither to ensure whether the trained LLM is logically consistent in these facts or not.

Table 18: Experimental results on MiniCheck

| Facts | Accuracy | | Consistency | |
|---|---|---|---|---|
| | MiniCheck-FT5 | Fine-tuned Llama2-7B | MiniCheck-FT5 | Fine-tuned Llama2-7B |
| $p, \neg p$ | 0.57 | **0.97** | 0.13 | **0.94** |
| $p \wedge q$ | 0.83 | **0.87** | 0.84 | **0.86** |
| $p \vee q$ | 0.39 | **0.81** | 0.41 | **0.77** |

## K  EXPERIMENTS WITH LARGER AND CLOSED-SOURCE MODELS

### K.1  LLAMA2-70B

In this section, we present experimental results on models with a larger number of parameters, such as Llama2-70B . The experiments on Llama2-70B are presented in Table 19, Table 20 and Table 21.

From Table 19 and Table 20, we observe that despite having more parameters than the corresponding 7B and 13B versions of LLaMA2, the Llama2-70B base model does not always demonstrate an improved logical consistency in all query types: simple facts, complex facts, and logic rules, demanding further supervised fine-tuning or instruction prompting. Fine-tuning of Llama2-70B is incomplete due to its higher demand on computational resources.

Llama2-70B demonstrates higher effectiveness in understanding instruction prompts than the lower-size models from the same family. For example, in Table 21, we observe that Llama2-70B improves accuracy from 0.77 to 0.99 and consistency from 0.53 to 0.99 due to instruction prompting, which is superior to Llama2-13B. As such, we hypothesize that fine-tuning may not be necessary on larger models, which already demand high computational resources; instruction prompting may be as competitive as fine-tuning.

### K.2  GPT-4O

In this section, we present experimental results on GPT-4o. Table 22 presents the results on simple facts and complex facts and logic rules on the Freebase dataset. In Table 23, we present the results on more complex logic rules such as the law of syllogism and first-order logic.

Table 19: Comparison among models of different sizes regarding accuracy and logical consistency of LLMs before and after fine-tuning (FT). Bold numbers denote an improved result in accuracy and consistency for fine-tuning. Training is performed on FreebaseLFC dataset only (marked with '∗'), while performance improved in all datasets. '—' denotes incomplete fine-tuning due to computational resources.

| Model | Dataset | Fact | Accuracy | | Logical Consistency | |
|---|---|---|---|---|---|---|
| | | | Before FT | After FT | Before FT | After FT |
| Llama2-70B | FreebaseLFC* | $p, \neg p$ | 0.77 | — | 0.53 | — |
| | | $p \wedge q$ | 0.91 | — | 0.86 | — |
| | | $p \vee q$ | 0.32 | — | 0.62 | — |
| Llama2-13B | FreebaseLFC* | $p, \neg p$ | 0.90 | **0.93** | 0.81 | **0.86** |
| | | $p \wedge q$ | 0.61 | **0.93** | 0.67 | **0.83** |
| | | $p \vee q$ | 0.73 | **0.76** | 0.73 | **0.97** |
| Llama2-7B | FreebaseLFC* | $p, \neg p$ | 0.87 | **0.97** | 0.76 | **0.94** |
| | | $p \wedge q$ | 0.39 | **0.87** | 0.47 | **0.86** |
| | | $p \vee q$ | 0.78 | **0.81** | 0.83 | 0.77 |
| Gemma-2B | FreebaseLFC* | $p, \neg p$ | 0.82 | **0.98** | 0.66 | **0.96** |
| | | $p \wedge q$ | 0.45 | **0.83** | 0.70 | **0.84** |
| | | $p \vee q$ | 0.73 | **0.94** | 0.91 | 0.90 |

Table 20: Comparison among models of various sizes regarding their generalizability to complex facts and rules.

| Model | Dataset | Fact | Accuracy | | Logical Consistency | |
|---|---|---|---|---|---|---|
| | | | Before FT | After FT | Before FT | After FT |
| Llama2-70B | FreebaseLFC | $p \vee (q \wedge r)$ | 0.38 | — | 0.70 | — |
| | | $p \wedge (q \vee r)$ | 0.70 | — | 0.90 | — |
| | | $p \vee q \leftrightarrow q \vee p$ | 0.32 | — | 0.94 | — |
| | | $p \wedge q \leftrightarrow q \wedge p$ | 0.91 | — | 0.93 | — |
| Llama2-13B | FreebaseLFC | $p \vee (q \wedge r)$ | 0.74 | **0.85** | 0.78 | **0.81** |
| | | $p \wedge (q \vee r)$ | 0.53 | **0.82** | 0.63 | **0.79** |
| | | $p \vee q \leftrightarrow q \vee p$ | 0.73 | **0.76** | 0.85 | **0.98** |
| | | $p \wedge q \leftrightarrow q \wedge p$ | 0.61 | **0.93** | 0.84 | **0.91** |
| Llama2-7B | FreebaseLFC | $p \vee (q \wedge r)$ | 0.68 | **0.74** | 0.78 | 0.71 |
| | | $p \wedge (q \vee r)$ | 0.42 | **0.74** | 0.54 | **0.75** |
| | | $p \vee q \leftrightarrow q \vee p$ | 0.78 | **0.84** | 0.87 | 0.85 |
| | | $p \wedge q \leftrightarrow q \wedge p$ | 0.39 | **0.89** | 0.91 | **0.92** |
| Gemma-2B | FreebaseLFC | $p \vee (q \wedge r)$ | 0.62 | **0.83** | 0.91 | 0.77 |
| | | $p \wedge (q \vee r)$ | 0.38 | **0.74** | 0.77 | **0.81** |
| | | $p \vee q \leftrightarrow q \vee p$ | 0.73 | **0.94** | 0.90 | **0.93** |
| | | $p \wedge q \leftrightarrow q \wedge p$ | 0.45 | **0.83** | 0.80 | **0.90** |

From table 22, we observe that GPT-4o demonstrates improved logical consistency on negation, conjunction, and disjunction facts, but performs comparatively poorly when multiple logical operators are involved, such as $p \wedge (q \vee r)$ and $p \vee (q \wedge r)$ - the accuracy and consistency reduce to (0.84, 0.81) and (0.76, 0.87), respectively. Considering the API cost of running GPT-4o, we consider a smaller number of queries (around 1000 queries per query type), resulting in a total cost of 45 US dollars.

The results in Table 23 suggest that GPT-4o lacks logical consistency on complex logical expressions or higher-order Horn rule settings, justifying the motivation of our work, that is, to measure and improve the logic consistency of LLMs.

Finally, we proceeded to the instruction prompt for GPT-4o to improve its consistency on simple facts. We find that the logical consistency improves to 0.98, as shown in table 24.

Table 21: Impact of instruction prompting (Prompt) vs. fine-tuning (FT) on simple facts across models of various sizes.

| Model | Dataset | Accuracy | | | Logical Consistency | | |
|-------|---------|----------|--------|----------|---------------------|--------|----------|
| | | Before FT | Prompt | After FT | Before FT | Prompt | After FT |
| Llama2-70B | FreebaseLFC | 0.77 | **0.99** | — | 0.53 | **0.99** | — |
| Llama2-13B | FreebaseLFC | 0.90 | 0.89 | **0.93** | 0.81 | 0.79 | **0.86** |
| Llama2-7B | FreebaseLFC | 0.87 | 0.82 | **0.97** | 0.76 | 0.64 | **0.94** |
| Gemma-2B | FreebaseLFC | 0.82 | 0.88 | **0.98** | 0.66 | 0.77 | **0.96** |

Table 22: Logical consistency and accuracy of GPT-4o model (before fine-tuning) across different facts and logic rules in FreebaseLFC benchmark.

| Fact | Accuracy | Logical Consistency |
|------|----------|---------------------|
| $p, \neg p$ | 0.91 | 0.91 |
| $p \wedge q$ | 0.97 | 0.95 |
| $p \vee q$ | 0.97 | 0.95 |
| $p \wedge q \leftrightarrow q \wedge p$ | 0.96 | 0.95 |
| $p \vee (q \wedge r)$ | 0.76 | 0.87 |
| $p \wedge (q \vee r)$ | 0.84 | 0.81 |

**Comparing GPT-4o with smaller LLMs.** We compare GPT-4o with smaller models such as Gemma-2B, Llama2-7B, and Llama2-13B on FoL and LoS queries in Table 23. We observe that the base 2B, 7B, and 13B models do not offer similar levels of accuracy and consistency on first-order logic. This is not surprising, since GPT-4o has been exposed to structured logic (Hurst et al., 2024). We also observe that Llama2-13B and Gemma-2B have better consistency than GPT-4o on the LoS dataset, but their accuracy is lacking in comparison. It is plausible for a (base) model to demonstrate higher consistency yet lower accuracy since consistency indicates the logical robustness of the model on logically manipulated queries independent from the matter of accuracy. In our attempt to further improve GPT-4o, we have conducted instruction prompting with simple facts. The results are shown in Table 24. This result indicates that since fine-tuning is often infeasible in large closed-source models like GPT-4o, instruction prompting becomes effective in improving both accuracy and consistency to 98%.

Table 23: Performance of the LLMs on more complex logic settings: first-order logic with an existential quantifier ($\exists$) and the law of syllogism curated from the FreebaseLFC benchmark.

| Fact | Models | Accuracy | Logical Consistency |
|------|--------|----------|---------------------|
| First-order Logic (FoL) | GPT-4o | 0.91 | 0.78 |
| | Llama2-13B | 0.57 | 0.13 |
| | Llama2-7B | 0.50 | 0.00 |
| | Gemma-2B | 0.50 | 0.00 |
| Law of Syllogism (LoS) | GPT-4o | 0.94 | 0.64 |
| | Llama2-13B | 0.68 | 0.73 |
| | Llama2-7B | 0.55 | 0.45 |
| | Gemma-2B | 0.55 | 0.76 |

Table 24: Impact of instruction prompting (Prompt) on GPT-4o in improving the accuracy and logical consistency of simple facts in FreebaseLFC benchmark.

| | Accuracy | | Logical Consistency | |
|---|---|---|---|---|
| | Before FT | Prompt | Before FT | Prompt |
| | 0.91 | **0.98** | 0.91 | **0.98** |

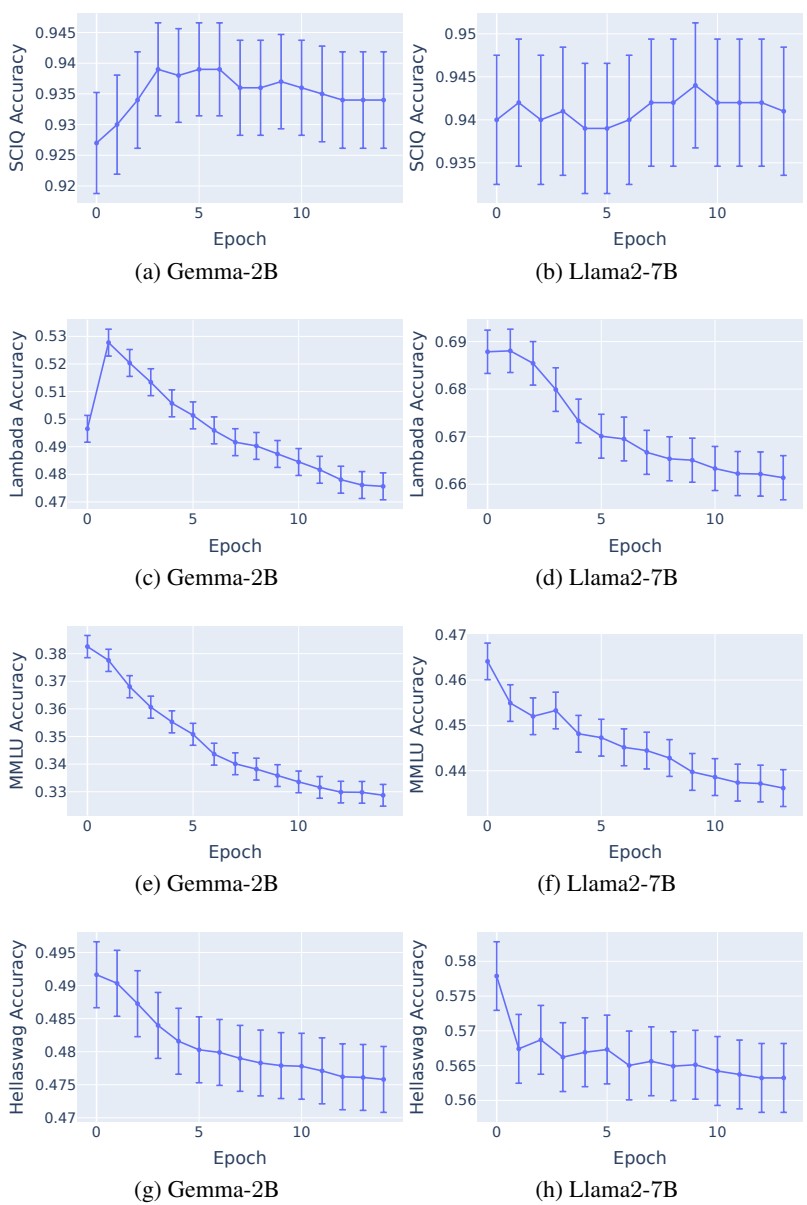

Figure 11: Performance on Language benchmarks

## L    PERFORMANCE ON LANGUAGE BENCHMARKS

We conduct an evaluation of the fine-tuned SFT model on four general language benchmarks: SCIQ (Welbl et al., 2017), Lambada (Paperno et al., 2016), MMLU (Hendrycks et al., 2020), and Hellaswag (Zellers et al., 2019). Figure 11 shows the performance across epochs.

We observe that the logical consistency-based fine-tuning improves the benchmark accuracy on SCIQ (from $0.926$ to $0.94$ in Gemma-2B and $0.94$ to $0.945$ in Llama2-7B), which is a science exam question-answer benchmark. In addition, on the Lambada benchmark, the performance increases in the initial few epochs. In the rest of the benchmarks, MMLU and Hellaswag, increasing fine-tuning epochs results in a monotonic decrease in benchmark accuracy with epochs.

Therefore, consistently across multiple models like Gemma and Llama, consistency-based fine-tuning is helpful in the scientific domain benchmark. At the same time, performance degradation may be observed in other multi-task abilities with excessive fine-tuning. Therefore, as evident from our experiments, we suggest applying consistency-focused SFT for a few epochs before the model overfits and loses its generalization performance on other benchmarks.

# M    ACKNOWLEDGMENT

AK and SH acknowledge support from the Novo Nordisk Foundation grant NNF 22OC0072415.

