# OpenReview forum: "Logical Consistency of Large Language Models in Fact-Checking"
_ICLR.cc/2025/Conference — ICLR 2025 Poster_

### Official Review · Reviewer_JETn · 2024-10-17

**Soundness:** 3
**Presentation:** 3
**Contribution:** 3
**Rating:** 6
**Confidence:** 4

**Summary:**

This paper focuses on the logical consistency of LLMs in fact-checking using KGs. The authors first formalize the definition of logical consistency for LLMs and concentrate on retrieval-augmented LLMs within a fact-checking task. They propose three benchmarks, FreebaseLFC, NELLLFC, and WikiLFC to demonstrate that several existing LLMs lack logical consistency when handling complex queries, and employ supervised fine-tuning to enhance the logical consistency of these models.

**Strengths:**

1. This paper is well-written and easy to follow.

2. The logical consistency evaluation of LLMs is rational and interesting.

3. The employed supervised fine-tuning method on LLMs has enhanced the logical consistency performance.

**Weaknesses:**

1. The authors may want to provide more results across various scales of LLMs (including both smaller models such as Vicuna-68M and larger models like LLaMA2-70B) to comprehensively evaluate the logical consistency issue.

2. The authors may want to include more results from existing fact-checking methods using KGs, such as MiniCheck [1] or RefChecker [2], on the proposed benchmarks.

[1] Minicheck: Efficient fact-checking of llms on grounding documents. EMNLP, 2024.


[2] Refchecker: Reference-based fine-grained hallucination checker and benchmark for large language models. 2024.

3. The third contribution, "Improvement of Logical Consistency," appears somewhat underwhelming. In Table 1, Vanilla LLaMA-2-13B already achieved strong results before fine-tuning, which leads me to wonder whether models like GPT-4o, GPT-o1 exhibit lower levels of logical inconsistency which may be hardly noticeable.

4. The authors may want to discuss and summarize more LLMs with KG contexts methods in related work including [3] to provide a more comprehensive insight.

[3] Coarse-to-Fine Highlighting: Reducing Knowledge Hallucination in Large Language Models. ICML2024.

**Questions:**

1. Have the authors considered more complex logical expressions or some higher-order Horn rules?

2. The tables in the paper are not formatted as three-line tables, which appears somewhat unusual.

---

> ### Author Response · Authors · 2024-11-20
> **Rebuttal**
>
> Thank you for appreciating the write-up and contributions of the paper. In the following, we incorporate additional experiments suggested by the reviewer and provide a summary of our findings. We will update the final version of the paper accordingly.
>
>
>
>
>
> > The authors may want to provide more results across various scales of LLMs (including both smaller models such as Vicuna-68M and larger models like LLaMA2-70B) to comprehensively evaluate the logical consistency issue.
>
> Thanks for the suggestion of extending our experiments to both smaller and larger models.
>
>
>
> Throughout our experiments, we consider the chat version of the base model like gemma-2b-it, llama2-7b-chat-hf (Section 5.1) because of the superior capability of chat models in question-answering. Since there is no corresponding chat version for Vicuna-68M, we anticipate its ability to understand factual questions might be limited. In our experiments during rebuttal, the base Vicuna-68M demonstrates a random accuracy of 0.5 and consistency of 0 on simple facts from both FreebaseLFC and FEVER datasets (containing natural text queries, as suggested by Reviewer pZwi). As such, the smallest model Vicuna-68M performs poorly in logical consistency.
>
>
>
> We experimented with llama2-70b-chat-hf, the chat version of LLaMA2-70B, where we provide detailed results in Appendix I in the updated version of the paper. In summary, despite having more parameters than the corresponding 7B and 13B versions of LLaMA2, the 70B base model does not always demonstrate an improved logical consistency in all query types: simple facts, complex facts, and logic rules, demanding further supervised fine-tuning or instruction prompting.
>
>
>
> We realize that fine-tuning LLaMA2-70B requires higher computational resources. In this context, LLaMA2-70B demonstrates higher effectiveness in understanding instruction prompts than the lower-size models from the same family. For example, in Table 19, LLaMA2-70B improves accuracy from 0.77 to 0.99 and consistency from 0.53 to 0.99 due to instruction prompting, which is superior to LLaMa2-13B. As such, we hypothesize that fine-tuning may not be necessary on larger models, which already demand high computational resources; instruction prompting may be as competitive as fine-tuning.
>
> > The authors may want to include more results from existing fact-checking methods using KGs, such as MiniCheck [1] or RefChecker [2], on the proposed benchmarks.
>
> We present the results by considering state-of-the-art fact-checking methods using KGs such as MiniCheck [1] below:
> |                     | **Accuracy  (Minicheck-FT5)** | **Accuracy  (Fine-tuned Llama2-7b)** | **Consistency  (Minicheck-FT5)**| **Consistency  (Fine-tuned Llama2-7b)** |
> |---------------------|----------------------------|-------------------------------|--------------------------------------|-----------------------------------------|
> | **$p, \neg p$** | 0.57                       |     **0.97**                      |  0.13                                | **0.94**                                    |
> | **$p \land q$**    | 0.83                       |     **0.87**                    |  0.84                                  | **0.86**                                    |
> | **$p \lor q$**    | 0.39                       |     **0.81**                     |   0.41                               | **0.77**                                    |
>
>
> MiniCheck performs poorly because it was designed for fact-checking on grounding documents, not for verifying propositional logic queries, unlike ours, or to ensure whether the trained LLM is logically consistent in these facts.
>
>
>
>
>
>
> > The third contribution, "Improvement of Logical Consistency," appears somewhat underwhelming. In Table 1, Vanilla LLaMA-2-13B already achieved strong results before fine-tuning, which leads me to wonder whether models like GPT-4o, GPT-o1 exhibit lower levels of logical inconsistency which may be hardly noticeable.
>
> We are currently running experiments on GPT-4o. We will present the results once they are available.

---

> ### Author Response · Authors · 2024-11-20
> **Rebuttal Continued**
>
> > The authors may want to discuss and summarize more LLMs with KG contexts methods in related work including [3] to provide a more comprehensive insight.
>
>
> LLM-Embedder [3] is a general-purpose retrieval technique that learns vector embedding by rewarding high-quality retrieval candidates, contrastive learning, and knowledge distillation. LLM-Embedder can replace the Sentence transformer in the BFS+Vector Embedding method (Appendix, Line 1124-1130) introduced in our work. Like LLM-Embedder, one can also employ other embedding methods, such as BGE [4] and LLM-R [5], to retrieve more accurate triplets and construct more relevant contexts.
>
>
> Since we have already shown that vector embedding methods improve logical consistency, we did not experiment with other vector embedding methods.
>
>
>
>
>
>
>
>
>
> > Have the authors considered more complex logical expressions or some higher-order Horn rules?
>
>
> We have considered complex logical expressions and rules, such as commutative and distributive rules, to assess the generalizability of the fine-tuned model. The results are presented in Table 2. We have not tested the performance on “more” complex logical expressions since Table 2 already points out that LLMs finetuned on negation, disjunction, and conjunction are generalizable to more complex logical expressions.
>
>
>
> We thank you for bringing up higher-order Horn clauses, such as first-order logic. Extending the current approach to first-order logic, which is more general than propositional logic (the focus of this paper), might be effective for multi-class logic answer consistency. For instance, consider the claim “<Crimean> <is a part of> <Russia>” (asked by Reviewer 7P9X). The answer is neither yes nor no, but it depends on who claims it. If there is an equal proportion of evidence/references that support and refute the entity “(<Crimean> <is a part of> <Russia>).” we might categorize it into a third class, such as “It depends.” in contrast to binary “true” / “false.”  However, such an extension will require a significant number of changes in our retrieval component, context creation, and the overall methodology of logical consistency evaluation. We realized that this exploration demands a separate work; hence, we leave it as a future work.
>
>
> > The tables in the paper are not formatted as three-line tables, which appears somewhat unusual.
>
>
>
> Thank you for your attention to this issue. We have formatted all the tables in the updated draft as three-line tables.
>
>
> ` References: `
>
>
>
>
> [1] Minicheck: Efficient fact-checking of llms on grounding documents. EMNLP, 2024.
>
>
> [2] Refchecker: Reference-based fine-grained hallucination checker and benchmark for large language models. 2024.
>
>
> [3] Coarse-to-Fine Highlighting: Reducing Knowledge Hallucination in Large Language Models. ICML2024.
>
>
> [4] BAAI General Embedding.
>
>
> [5] Learning to Retrieve In-Context Examples for Large Language Models

---

> > ### Author Response · Authors · 2024-11-22
> > **Results with GPT-4o**
> >
> > We experiment with GPT-4o on simple and complex facts, and logic rules from the FreebaseLFC benchmark. The results are shown below. GPT-4o demonstrates improved logical consistency on negation, conjunction, and disjunction facts, but performs comparatively poorly when multiple logical operators are involved, such as $p \wedge (q \vee r)$ and $p \vee (q \wedge r)$ - the accuracy and consistency lower to (0.84, 0.81) and (0.76, 0.87), respectively. Considering the API cost of running GPT-4o, we consider a smaller number of queries (around 1000 queries per query type), resulting in a total cost of 38 US dollars.
> >
> > *Accuracy and Logical Consistency of GPT-4o*
> >
> > |  | **Accuracy** | **Consistency** |
> > |---|---|---|
> > | **$p, \neg p$** | 0.91 | 0.91 |
> > | **$p \land q$** | 0.97 | 0.95 |
> > | **$p \lor q$** | 0.97 | 0.95 |
> > | **$p \land q \leftrightarrow q \land p$** | 0.96 | 0.95 |
> > | **$p \wedge (q \vee r)$** | 0.84 | 0.81 |
> > | **$p \vee (q \wedge r)$** | 0.76 | 0.87 |
> >
> > Albeit having better consistency than Llama and Gemma model families, GPT-4o is not fully consistent. For example, on simple facts  $p, \neg p$, the logical consistency is 0.91. Similar to other LLMs, we proceed to the instruction prompt for GPT-4o (Section 4.1) and find that the logical consistency improves to 0.98, as shown in the table below.
> >
> >
> > |  | **Accuracy** | **Accuracy (Instruction Prompting)** | **Consistency** | **Consistency (Instruction Prompting)** |
> > |---|---|---|---|---|
> > | **$p, \neg p$** | 0.91 | **0.98** | 0.91 | **0.98**
> >
> >
> >
> >
> > Since GPT-4o is closed source, it is infeasible to integrate our RAG component during training, and perform our SFT. Hence we are unable to fine-tune GPT-4o in the way we fine-tuned other models, e.g. Llama and Gemma model families. From this experiment and from the experiment with Llama2-70B (suggested by Reviewer JETn), it seems instruction prompting on larger LLMs is as competitive as SFT on smaller LLMs.
> >
> >
> > In conclusion, our paper highlights the importance of logical consistency of an LLM (both open-source and closed source); and we demonstrate that a consistency focused instruction prompting (effective in large models) and fine-tuning (effective in small models) can improve their logical consistency.

---

> ### Author Response · Authors · 2024-11-23
> **Looking forward to hearing from you!**
>
> Dear Reviewer JETn,
>
> We wonder whether we can provide any additional information to clarify your questions/concerns about additional experiments, the necessity of our third contribution, and the extension to more complex logical expressions.
>
> Thank you for your constructive comments; we look forward to your feedback!
>
> Paper 3993 Authors

---

> ### Comment · Reviewer_JETn · 2024-11-24
>
> I thank the authors for their responses.
>
> 1. While I agree that logical consistency is a consideration in constructing trustworthy LLMs, the GPT-4o results—where the majority of judgments performed exceptionally well—suggest that it might be more challenging to observe logical inconsistency issues with o1 under the current simple rule settings. I strongly encourage the authors to consider the use of more complex logical expressions or higher-order Horn rule settings when updating the dataset. Although the basic logical operators may be complete, practical implementation often requires complex and potentially redundant logical relations.
>
> 2. Additionally, the updated PDF does not seem to discuss potentially relevant papers on LLM hallucinations.
>
> Based on these considerations, **I am inclined to maintain the current scores for now.**  I am expecting that the authors consider incorporating more complex logical expressions or higher-order Horn rule settings.

---

> > ### Author Response · Authors · 2024-11-25
> >
> > Thank you for reading and responding to our rebuttal.
> >
> >
> > > 1. While I agree that logical consistency is a consideration in constructing trustworthy LLMs, the GPT-4o results—where the majority of judgments performed exceptionally well—suggest that it might be more challenging to observe logical inconsistency issues with o1 under the current simple rule settings. I strongly encourage the authors to consider the use of more complex logical expressions or higher-order Horn rule settings when updating the dataset. Although the basic logical operators may be complete, practical implementation often requires complex and potentially redundant logical relations.
> >
> >
> >
> >
> > Following your recommendation, we have updated our benchmark by curating two more datasets from Freebase containing more complex logical expressions, especially the law of Syllogism (LoS) and the first-order logic (FoL).
> >
> >
> >
> >
> > In the LoS dataset, there are 100 queries of the form $r_1 (p, q) \land r_2(q, s)$ encapsulating the logic $(p \Rightarrow q) \land (q \Rightarrow s)$, which, according to the law of Syllogism, should be equivalent to $p \implies s$. An example: $Contains(Oceania, New Zealand) \land Country(New Zealand, Auckland)$ is logically equivalent to $Contains(Oceania, Auckland)$.
> >
> >
> > In the FoL dataset, there are 500 queries of the form $\exists_x r_1(p, x) \land r_2(x,q)$ and their negations in the form of $\forall_x \neg r_1(p,x) \lor \neg r_2(x,q)$. The negation of the former should be logically equivalent to the latter.
> >
> >
> >
> >
> > We have conducted experiments on GPT-4o and present the results below:
> >
> >
> >
> >
> > |  | **Accuracy** | **Consistency** |
> > |---|---|---|
> > | **LoS** | 0.94 | 0.64 |
> > | **FoL** | 0.91 | 0.78 |
> >
> >
> >
> >
> > These results suggest that **GPT-4o lacks logical consistency on complex logical expressions or higher-order Horn rule settings**, justifying the motivation of our work, that is, to measure and improve the logic consistency of LLMs.
> >
> >
> > *We have updated the revised manuscript in Section I of the Appendix to include the results with GPT-4o.*
> >
> >
> > > 2. Additionally, the updated PDF does not seem to discuss potentially relevant papers on LLM hallucinations.
> >
> >
> > We have added section B.3  (Large Language Models Hallucinations) in the Appendix, where we discussed the potentially relevant papers such as [1][2][3][4][5] on LLM hallucinations that affect consistency.
> >
> >
> > [1] Coarse-to-Fine Highlighting: Reducing Knowledge Hallucination in Large Language Models. ICML2024.
> >
> >
> > [2] A survey on hallucination in large language models: principles, taxonomy, challenges, and open questions, ACM Transactions on Information Systems 2024.
> >
> >
> > [3] Self-consistency improves chain of thought reasoning in language models, ICLR 2023.
> >
> >
> > [4] Learning to retrieve in-context examples for large language models. In EACL, pp. 1752–1767, 2024
> >
> >
> > [5] Chain-of-thought prompting elicits reasoning in large language models. In NeurIPS, 2022.
> >
> >
> >
> >
> >
> >
> > --------
> > **Thank you again for your comments! We hope to have addressed your concerns. We would be happy to answer any follow-up questions or comments!**

---

> ### Comment · Reviewer_JETn · 2024-11-26
>
> I appreciate the authors for providing more complex Horn rule cases as well as the additional results for tesing GPT-4o. I strongly suggest including these rule conbinations or even more complex rule combiantions into the dataset. Additionally, I am curious about the performance of these complex Horn rules on models of 2B, 7B, and 13B scales (as reported in the main text), and whether fine-tuning some 7B models might lead to improvements. **Given that there is additional discussion time, I look forward to this set of results being used to reconsider my current rating scores.**

---

> > ### Author Response · Authors · 2024-11-28
> >
> > Thank you for engaging with us with your constructive comments and feedback.
> >
> >
> >
> > > I strongly suggest including these rule conbinations or even more complex rule combiantions into the dataset. Additionally, I am curious about the performance of these complex Horn rules on models of 2B, 7B, and 13B scales (as reported in the main text), and whether fine-tuning some 7B models might lead to improvements.
> >
> >
> >
> > We have included these rule combinations into our dataset. In particular, we have evaluated the Llama2-13B, Llama2-7B, and Gemma-2B base models with GPT-4o in the following table and in Table 22 of the revised paper. Since SFT takes time, we are still conducting fine-tuning with this updated dataset (combination of propositional logic with FoL + LoS). We will respond with these results when they are available.
> >
> >
> > From the table below (also Table 22 in the revised paper) we observe that the base 2B, 7B, and 13B models do not offer similar levels of accuracy and consistency on first-order logic. This is not surprising, since GPT-4o has been exposed to structured logic [1]. We also observe that Llama2-13B and Gemma-2B have better consistency than GPT-4o on the LoS dataset, but their accuracy is lacking in comparison.  It is plausible for a (base) model to demonstrate higher consistency yet lower accuracy, since consistency indicates the logical robustness of the model on logically manipulated queries regardless of being accurate. Further instruction prompting and/or fine-tuning are therefore necessary to be both accurate and consistent, as demonstrated in Section 5.
> >
> >
> >
> >
> > | **Facts** | **Models** | **Accuracy** | **Consistency** |
> > |---|---|---|---|
> > | **First-order Logic (FoL)** | GPT-4o  | 0.91  | 0.78 |
> > |  | Llama2-13B  | 0.57  | 0.13 |
> > |  | Llama2-7B  | 0.50  | 0.00 |
> > |  | Gemma-2B  | 0.50  | 0.00 |
> > | **Law of Syllogism (LoS)** | GPT-4o  | 0.94  | 0.64 |
> > |  | Llama2-13B  | 0.68  | 0.73 |
> > |  | Llama2-7B  | 0.55  | 0.45 |
> > |  | Gemma-2B | 0.55  | 0.76 |
> >
> >
> > [1] GPT-4o System Card. OpenAI. ( https://cdn.openai.com/gpt-4o-system-card.pdf )

---

> ### Comment · Reviewer_JETn · 2024-11-28
>
> Thank you for the author's response. I greatly appreciate the author's willingness to incorporate these more complex Horn rule settings. The inconsistencies observed in the model without special training further highlight the necessity of logical consistency. I look forward to the results after SFT. **I have raised my score to 6.** I also increased soundness and contribution scores accordingly.

---

> > ### Author Response · Authors · 2024-12-01
> >
> > Thank you for increasing your score!
> >
> >
> >
> > We have evaluated the performance of Fine-tuned LLMs on First-order logic and Law of Syllogism queries and present the results in the table below.
> >
> >
> >
> >
> >
> > The column denoted as “After FT” indicates the performance of the LLMs that have been fine-tuned based on only simple and complex facts from FreebaseLFC (similar to Table 1). The column denoted as “After FT w/ FoL + LoS” indicates the performance of the LLMs that have been fine-tuned based on simple, complex, FoL and LoS facts from FreebaseLFC. We produced this new fine-tuned model to verify if knowledge about some FoL and LoS queries can further improve the LLMs' consistency and accuracy.
> >
> >
> >
> >
> > | **Facts** | **Model** | **Accuracy (Before FT)** | **Accuracy (After FT)** | **Accuracy (After FT w/ FoL + LoS)** | **Consistency (Before FT)** | **Consistency (After FT)** | **Consistency (After FT w/ FoL+LoS)** |
> > |---|---|---|---|---|---|---|---|
> > | **First-order Logic (FoL)** | Llama2-7B  | 0.50  | 0.61 | **0.91** | 0.00 | 0.45 | **0.86** |
> > | **Law of Syllogism (LoS)** | Llama2-7B  | 0.55  | 0.77 | **0.98** | 0.45 | 0.55 | **0.97** |
> >
> >
> >
> >
> >
> > Therefore, fine-tuning improves accuracy and consistency in both FoL and LoS queries. Once again, this highlights the necessity of SFT, as we have shown in the paper.
> >
> >
> >
> > We shall add these new sets of results in the final version of our paper.
> >
> >
> >
> >
> >
> > Thank you again for your valuable input which helped us improve our paper. *We hope this new set of experiments will increase your support for our paper*.

---

### Official Review · Reviewer_7P9X · 2024-11-01

**Soundness:** 3
**Presentation:** 3
**Contribution:** 3
**Rating:** 6
**Confidence:** 3

**Summary:**

This paper tackles the significant issue of logical consistency in responses generated by large language models (LLMs). It emphasizes a gap in current research, where consistency assessments focus primarily on simple paraphrasing rather than complex logical reasoning. The authors address this by examining how LLMs handle propositional logic (involving operators such as negation, conjunction, and disjunction) within a fact-checking framework supported by knowledge graphs (KGs).

**Strengths:**

There are three main contributions of this paper.
1) Novel Dataset and Benchmarking: Three logical fact-checking datasets FreebaseLFC, NELLLFC, and WikiLFC, derived from knowledge graphs.
2) Evaluating LLMs on these new fact-checking datasets. The paper includes a variety of experimental setups, including comparisons of zero-shot instruction prompting and supervised fine-tuning, adding depth to the evaluation of LLMs' logical consistency. Results show that existing LLMs are not consistent when considering
logical equivalents or variants of these facts.
3) Improving the consistency of LLMs via supervised fine-tuning

**Weaknesses:**

1)  While the paper presents methods for extracting relevant KG context through BFS and embedding-based retrieval, it is unclear how effective these methods are in dynamically varying real-world contexts. Expanding this section could strengthen the applicability of the approach.
2) The paper suggests fine-tuning as a solution to improve logical consistency but does not deeply discuss the associated computational overheads and limitations, which may impact the scalability of this approach for large datasets or diverse domains.
3) The paper mainly focuses on binary responses, which may limit the generalizability of the method in complex domains. Exploring extensions to multi-class logic and non-binary answer consistency could open avenues for broader applications. Note that a fact might not be binary, e.g., Crimean is a part of Russian. The answer is nether yes or no, but depends on who claims it.

**Questions:**

See Weaknesses.

---

> ### Author Response · Authors · 2024-11-20
> **Rebuttal**
>
> We thank the reviewer for the valuable feedback and for suggesting interesting future directions to explore. In the following, we answer specific questions.
>
>
>
>
>
> > While the paper presents methods for extracting relevant KG context through BFS and embedding-based retrieval, it is unclear how effective these methods are in dynamically varying real-world contexts. Expanding this section could strengthen the applicability of the approach.
>
> Thank you for this excellent question! Indeed, in practice, real-world applications may need context retrieval to quickly adapt to new information (e.g., the knowledge graph evolving with time) or shifting user needs (e.g., users wanting to adapt their LLM fine-tuned for simple facts to be consistent w.r.t. more complex facts and logic rules).
>
>
>
> In the case of evolving KGs, the “BFS + Relation hierarchy” retrieval is a computationally better alternative than embedding-based retrieval methods such as the “BFS + Vector Embedding” or the “Vector Embedding (without BFS)”  method. Although the embedding-based methods slightly improve the accuracy and consistency (as shown in Table 11, Appendix E), due to their higher pre-processing overhead, they must be conducted offline repeatedly as the KG changes to obtain better entity embeddings.
>
>
>
> In the case of a shift in query needs, we considered such a case in section 5.2 (Generalizability of Fine-tuned Models), where we trained LLMs on simple facts and tested how they perform when the queries are out-of-distribution (complex facts and rules). Table 2 and Table 13 suggest that the proposed method still performs well in the case of such a shift in the query distribution.
>
>
>
> *As recommended, we have added a subsection in the Appendix discussing these cases (Page 22, “dynamically varying contexts”).*
>
>
>
> > The paper suggests fine-tuning as a solution to improve logical consistency but does not deeply discuss the associated computational overheads and limitations, which may impact the scalability of this approach for large datasets or diverse domains.
>
> **Fine-tuning Efficiency.** The associated computational overheads of fine-tuning are discussed in Appendix E (Efficiency). We compare the per-epoch running time of PEFT vs. full fine-tuning of Llama2-7B in Table 6. PEFT requires on average 4.43 hours to complete an epoch including the computation of evaluation loss, whereas full fine-tuning requires 12.52 hours. Therefore, PEFT is more efficient than full fine-tuning in improving the efficiency of supervised fine-tuning for logical consistency.
>
>
>
> *Alternate to Fine-tuning.* We also explore less computationally demanding approaches than fine-tuning, such as instruction prompting, few-shot prompting, and chain-of-thought prompting (Table 12 in the Appendix). For example, during rebuttal to Reviewer JETn, we find that small and moderate-size models such as Llama2-7B and Llama2-13B might be better off with fine-tuning, whereas higher size models like Llama2-70B are better off using instruction prompting in order to be logically consistent.
>
> **Inference Efficiency.** Correspondingly, Table 7 shows the end-to-end inference time for facts with different numbers of operators. We find that  the end-to-end fact-checking time is quite efficient (less than 0.63 seconds per sample) in our implementation.
>
> Regarding the running time of end-to-end fact-checking, the BFS-based context finding from the KG and LLM inference are parts of it. BFS time is correlated with the sparsity of the KG and can be approximated by how many edges are extracted in the BFS subgraph – for simple facts, the median number of edges in the BFS subgraph is 9773, 763.5, and 7 for FreebaseLFC, NELLLFC, and WikiLFC, respectively. Notice that analogously, in Table 7, the BFS time for FreebaseLFC is higher than that for NELLLFC, and WikiLFC has the lowest BFS time. Further, since WikiLFC is the sparsest among all KGs in our experiments, the context creation time and subsequent LLM inference time are lower in WikiLFC compared to the other two datasets – for example, the LLM inference time is the lowest in WikiLFC (0.08 seconds) compared to FreebaseLFC and NELLLFC (0.20 seconds).
>
>
> **Diverse domains.** Our approach is not necessarily reliant on any specific domain knowledge; rather, we propose a general-purpose technique to enhance the logical consistency of LLMs designed for any domain. For example, in the context of fact-checking, we fine-tune on FreebaseLFC, but demonstrate generalization on other two datasets: NELLLFC and WikiLFC. However, domain adaptation of logical consistency beyond fact-checking would be an interesting question to explore in the future.

---

> ### Author Response · Authors · 2024-11-20
> **Rebuttal Continued**
>
> > The paper mainly focuses on binary responses, which may limit the generalizability of the method in complex domains. Exploring extensions to multi-class logic and non-binary answer consistency could open avenues for broader applications. Note that a fact might not be binary, e.g., Crimean is a part of Russian. The answer is neither yes or no, but it depends on who claims it.
>
> Thank you for this insightful question. Handling multi-class logic and non-binary answer consistency would open avenues for broader applications. However, both are outside the scope of our current paper.
>
>
>
> Extending the current approach to first-order logic, which is more general than propositional logic (the focus of this paper), might be effective for multi-class logic answer consistency. For instance, if there is an equal proportion of evidence/references that support and refute the entity “(<Crimean> <is a part of> <Russia>).” we might categorize it into a third class such as “It depends.”, in contrast to binary “true” / “false.”  However, such an extension will require a significant number of changes in our retrieval component, context creation, and the overall methodology of logical consistency evaluation. We believe this exploration, by itself, demands a separate paper; hence, we leave it as a future work (Section 6).

---

> ### Author Response · Authors · 2024-11-23
> **Looking forward to hearing from you!**
>
> Dear Reviewer 7P9X,
>
> We wonder whether we can provide any additional information to clarify your questions/concerns about dynamic KG, computational overheads, scalability, and the method's generalizability to complex domains.
>
> Again, thank you very much for your constructive comments; we look forward to your feedback!
>
> Paper 3993 Authors

---

> ### Comment · Area_Chair_CGA1 · 2024-11-25
> **Reminder: Rebuttal Deadline for ICLR 2025**
>
> Dear Reviewer 7P9X,
>
> As the rebuttal deadline approaches, please kindly check the papers' discussion threads and respond to the authors' rebuttals. If you haven't had a chance to respond yet, I’d greatly appreciate your input soon. Your insights are invaluable to the authors and the review process.
>
> Thank you for your effort and support!
>
> Best regards,
>
> Area chair

---

> > ### Comment · Reviewer_7P9X · 2024-11-29
> > **Thank you**
> >
> > Thank you for addressing my concerns. After reading all reviews and rebuttals. I kept my score as weak accpet.

---

### Official Review · Reviewer_Eq6F · 2024-11-04

**Soundness:** 3
**Presentation:** 3
**Contribution:** 3
**Rating:** 5
**Confidence:** 3

**Summary:**

This paper discusses the topic of the logical consistency of LLM when answering complex queries that can be expressed by propositional logic and knowledge extracted from the knowledge graph. The logical consistency is defined by whether the model can answer the logical equivalent query with the same answer. To this end, the author first showcases some examples of non-consistency in query answering of LLM and then uses finetuning instead of prompting to improve the logical consistency of LLM.

**Strengths:**

1.	The definition of logical consistency is strict and the paper constructs three new datasets for finetuning and evaluating the consistency of LLM.
2.	The construction of the LLMQUERY seems interesting and the author proposes many optimizations towards that.
3.	The experiments show that the logical consistency can be improved by finetuning but not prompting.

**Weaknesses:**

1.	Though being logical consistent is a desirable attribute of LLM, we are also interested in whether it retrieve the correct answer. The logical consistency is only computed by whether the model gives the consistent answer with itself, but whether the answers are true or false are neglected in the experiment.
2.	The writing of this paper can be further improved. For example, the Proposition 3 is trivial and kind of redundant. There are also some mistakes like in Line 346 ``more computationally cheaper’’.

**Questions:**

Similar with weakness, I wonder whether the logical consistency helps with the reasoning ability of LLM?

---

> ### Author Response · Authors · 2024-11-20
> **Rebuttal**
>
> Thank you for your attention to the details of the paper and the questions. Below, we clarify the questions and address all comments in the updated version of the paper.
>
>
> > Though being logical consistent is a desirable attribute of LLM, we are also interested in whether it retrieve the correct answer. The logical consistency is only computed by whether the model gives the consistent answer with itself, but whether the answers are true or false are neglected in the experiment.
>
> Our definition of logical consistency is independent of whether the LLM retrieves the answer correctly or incorrectly. This is because we aim to separate the consistency measure from the accuracy measure. While achieving both is desirable, the end objectives are different. More specifically, accuracy focuses on factual correctness in fact-checking, whereas consistency evaluates the robustness of the LLM output under the logical perturbation of the target query. As such, if an LLM is accurate on a fact and is also consistent under logical manipulation of the same fact, it adds confidence to the end user that the LLM indeed understands the question while answering, which we highlight in the introduction (lines 74 to 76). However, even if the LLM is incorrect on a fact but logically consistent, it still provides confidence that the LLM may be factually incorrect on the specific fact but not an inconsistent (or flaky) one.
>
>
> Regardless of the separation between consistency and accuracy, we realize the importance of achieving both by an LLM. As such, our supervised fine-tuning technique relies on improving the accuracy of factual queries as a sufficient condition for achieving logical consistency (Proposition 2). *Furthermore, we report both the accuracy and logical consistency of LLMs in our experiments, where the proposed fine-tuning improves both. Specifically, in most of the cases in Table 1, whenever we observe increased accuracy, we observe improved consistency as well.*
>
>
>
>
>
>
>
>
> > The writing of this paper can be further improved. For example, Proposition 3 is trivial and kind of redundant.
>
> We assume the reviewer refers to Proposition 2 (not 3), since there is no Proposition 3 in the paper.
>
> The objective of Proposition 2 is to formalize how we aim to achieve logical consistency of an LLM by being accurate. More specifically, Proposition 2(i) proves that ‘being accurate’ on both a simple fact and its negation is a sufficient condition for an LLM to be logically consistent on simple facts. Similarly, Proposition 2(ii) proves that ‘being accurate’ on constituent atomic facts is sufficient for an LLM to be logically consistent on complex facts. Since our supervised fine-tuning is based on these two intuitive results, we have mentioned them in Section 4.2 and deferred the proof to the Appendix. Does the reviewer still think the proposition is redundant?
>
> > There are also some mistakes like in Line 346 ``more computationally cheaper’’.
>
> Thank you for pointing out this grammatical mistake in line 346. We have corrected it: “more computationally cheaper” => “computationally cheaper.”
>
>
> > Similar to weakness, I wonder whether logical consistency helps with the reasoning ability of LLM?
>
>
> Thank you for this thought-provoking question.
> There are many types of reasoning, such as causal reasoning, mathematical reasoning, abductive reasoning, and general reasoning for various NLP tasks, including question answering, natural language inference (NLI), and commonsense reasoning [1][2]. Although we do not focus on these types of reasoning in this paper, we have strong reasons to believe that logical consistency should help improve the LLMs' capability in these types of reasoning. This is simply because logical reasoning is the most basic type of reasoning that Humans apply to reason about causality and mathematics.
>
>
> LLM's general reasoning ability has been under much scrutiny in recent years, and experiments suggest that there is still a gap between LLMs and human-like logical reasoning ability [3]. Given such a state of affairs, it would be an interesting future study to evaluate how much logical consistency helps improve the general reasoning ability of LLMs.
>
> *We add the discussion in Appendix A (Limitations and broader impacts).*
>
> ` References: `
>
>
> [1] Trustworthy LLMs: a Survey and Guideline for Evaluating Large Language Models' Alignment
>
>
> [2] On the paradox of learning to reason from data.
>
>
> [3] True Detective: A Deep Abductive Reasoning Benchmark Undoable for GPT-3 and Challenging for GPT-4.

---

> ### Author Response · Authors · 2024-11-23
> **Looking forward to hearing from you!**
>
> Dear Reviewer Eq6F,
>
> We wonder whether we can provide any additional information to clarify your questions/concerns about logical consistency, proposition 2, and the paper write-up.
>
> Thank you again for your constructive comments; we are looking forward to your feedback!
>
> Paper 3993 Authors

---

> ### Comment · Area_Chair_CGA1 · 2024-11-25
> **Reminder: Rebuttal Deadline for ICLR 2025**
>
> Dear Reviewer Eq6F,
>
> As the rebuttal deadline approaches, please kindly check the papers' discussion threads and respond to the authors' rebuttals. If you haven't had a chance to respond yet, I’d greatly appreciate your input soon. Your insights are invaluable to the authors and the review process.
>
> Thank you for your effort and support!
>
> Best regards,
>
> Area chair

---

> > ### Comment · Reviewer_Eq6F · 2024-11-29
> >
> > Dear authors:
> >      Thank you for your rebuttal and I have read them carefully. However, there are still some concerns about that.
> >
> > >  Furthermore, we report both the accuracy and logical consistency of LLMs in our experiments, where the proposed fine-tuning improves both. Specifically, in most of the cases in Table 1, whenever we observe increased accuracy, we observe improved consistency as well.
> >
> > In the Table 1, we can only observe the result of p, $\lnot p$ together, it could be more convincing if the author can provide their separate results.
> >
> > > The objective of Proposition 2 is to formalize how we aim to achieve logical consistency of an LLM by being accurate.
> >
> > Thank you for your explanation, however, I also wonder why the author just chose DNF, rather than CNF or other canonical normal forms. I also wonder whether the LLM can be consistent with the De Morgan's law.

---

> ### Author Response · Authors · 2024-12-01
>
> Thank you for engaging with us with your constructive comments and feedback.
>
>
> > In the Table 1, we can only observe the result of p, $\lnot p$ together, it could be more convincing if the author can provide their separate results.
>
>
> To compute the logical consistency of a simple fact $p$, we consider a corresponding negative fact $\neg p$. We discuss two ways to negate a fact in the context of a knowledge graph (line 209 to 213 in Section 3.1). Since our goal is to assess consistency, which requires both positive and negative facts, we report their combined average accuracy in Table 1. Below, we separately report the accuracy of positive and negative facts and their consistency value before and after fine-tuning. Intuitively, the average of column 3 and 5 in the table below is the same as column 4 in Table 1 in the paper. In the final version, we will add the following table in the Appendix.
>
>
>
> |  |  | **Accuracy ($p$)** | **Accuracy ($p$)** | **Accuracy ($\neg p$)** | **Accuracy ($\neg p$)** | **Logical Consistency** | **Logical Consistency** |
> |---|---|:---:|---:|:---:|---:|:---:|---:|
> | **Model** | **Dataset** | **Before FT** | **After FT** | **Before FT** | **After FT** | **Before FT** | **After FT** |
> | Llama2-13B | FreebaseLFC | 0.86 | **1.00** | 0.95 | 0.86 | 0.81 | **0.86** |
> |  | NELLLFC | 0.88 | **0.99** | 0.87 | **0.94** | 0.76 | **0.93** |
> |  | WikiLFC | 0.95 | **1.00** | 0.97 | 0.93 | 0.92 | **0.93** |
> | Llama2-7B | FreebaseLFC | 0.91 | **0.97** | 0.83 | **0.97** | 0.76 | **0.94** |
> |  | NELLLFC | 0.97 | 0.94 | 0.46 | **0.94** | 0.44 | **0.88** |
> |  | WikiLFC | 0.98 | **0.99** | 0.81 | **0.82** | 0.80 | **0.81** |
> | Gemma-2B | FreebaseLFC | 0.92 | **1.00** | 0.73 | **0.96** | 0.66 | **0.96** |
> |  | NELLLFC | 0.93 | 0.93 | 0.68 | **0.96** | 0.62 | **0.89** |
> |  | WikiLFC | 0.99 | 0.98 | 0.52 | **0.82** | 0.52 | **0.80** |
>
>
>
> > Thank you for your explanation, however, I also wonder why the author just chose DNF, rather than CNF or other canonical normal forms.
>
>
> The choice of DNF in Proposition 1 is not the only design choice, since the proposed method for assessing logical consistency is not reliant on any specific normal form. In fact, the definition of consistency is flexible enough to be adapted to other canonical forms such as CNF. Informally, the consistency of the LLM response on a normal form query imposes an equivalence by applying respective logical operations on the LLM responses to constituent atomic queries. Below we present the adaptation of Proposition 1 to CNF with proof.
>
>
>
> **Adaptation of Proposition 1 to CNF.**
> An LLM is consistent with respect to a CNF fact $q=\wedge_{i=1}^n c_i$, where $c_i = \vee_{j=1}^{i_m} e_{ij}$,  if
>  \begin{equation*}
>     \mathtt{LLM}(q) = \bigwedge_{i=1}^{n} \left(\bigvee_{j=1}^{i_m} \mathtt{LLM}(e_{ij})\right).
>    \end{equation*}
> Here, $e_{ij}$ is an atomic relation fact for any $1\leq i\leq n
> $ and $1 \leq j \leq {i_m}$.
>
>
> **Proof.**
>  Applying the definition of consistency to the LLM response to $q$, we obtain
>
>
>
> $$
> \begin{align}
>         \mathtt{LLM}(q) &= \mathtt{LLM}(c_1) \wedge \mathtt{LLM}(c_2) \wedge \ldots \wedge \mathtt{LLM}(c_n) \\\\
>             & = \mathtt{LLM}( \bigvee_{j=1}^{1_m} e_{1j}) \wedge
>              \mathtt{LLM}( \bigvee_{j=1}^{2_m} e_{2j}) \wedge \ldots \wedge \mathtt{LLM}( \bigvee_{j=1}^{n_m} e_{nj}) \\\\
>         & = \left(\bigvee_{j=1}^{1_m} \mathtt{LLM}(e_{1j})\right) \wedge
>              \left(\bigvee_{j=1}^{2_m} \mathtt{LLM}(e_{2j})\right) \wedge \ldots \wedge \left(\bigvee_{j=1}^{n_m} \mathtt{LLM}(e_{nj})\right) \\\\
>         & = \bigwedge_{i=1}^{n} \left(\bigvee_{j=1}^{i_m} \mathtt{LLM}(e_{ij})\right)
> \end{align}
> $$
>
> In the final version of the paper, we shall highlight the generality of our consistency assessment to any normal form.
>
> > I also wonder whether the LLM can be consistent with De Morgan's law.
>
>
> The logical consistency assessment can be extended to De Morgan’s law. For example, for a conjunctive fact of two sub-queries $p \wedge q$, an LLM is logically consistent in De Morgan’s law if $\mathtt{LLM}(\neg (p \wedge q)) = \mathtt{LLM}(\neg p \vee \neg q)$. Similarly, for a disjunctive fact $p \vee q$, the LLM is logically consistent in De Morgan’s law if $\mathtt{LLM}(\neg (p \vee q)) = \mathtt{LLM}(\neg p \wedge \neg q)$.
>
>
>
>
>
> To be consistent in De Morgan’s law, the LLM requires the understanding of all three primitive operators $\neg, \wedge, \vee$. Table 2 and Table 13 in Appendix E suggest that the knowledge of $\neg, \wedge, \vee$ improves the consistency of LLMs in terms of more complex rules such as distributive law and commutative law which comprises these three primitive operators. Hence, *we expect that our method can improve the consistency with De Morgan’s law as well.*
>
>
>
>
>
>
>
>
>
> Thank you again for your valuable input which helped us improve our paper. *We hope this new set of explanations and details will increase your support for our paper.*

---

> ### Author Response · Authors · 2024-12-03
>
> Following the reviewer’s suggestion, we have fine-tuned Llama2-13B model on facts in logical forms involving three operators $ \neg $, $ \wedge $, and $ \vee $, and compare accuracy and logical consistency of De Morgan’s law before and after supervised fine-tuning. The key idea in fine-tuning is based on proposition 2, where we aim to achieve logical consistency by being accurate.
>
>
> |  | **Accuracy  (Llama2-13B)** | **Accuracy  (Fine-tuned Llama2-13B)** | **Consistency  (Llama2-13B)** |  **Consistency  (Fine-tuned Llama2-13B)** |
> |---|--:|--:|--:|--:|
> | $ \neg (p \vee q) \leftrightarrow \neg p \wedge \neg q $  |  0.25 | **0.99** |  1.00 | 0.98  |
> | $ \neg (p \wedge q) \leftrightarrow \neg p \vee \neg q $  | 0.75 | **0.97** | 0.99 | 0.95 |
>
>
> We observe that before any fine-tuning, the accuracy of De-Morgan’s law on conjunctive ($ \neg (p \wedge q) \leftrightarrow \neg p \vee \neg q$)  and disjunctive ($ \neg (p \vee q) \leftrightarrow \neg p \wedge \neg q $) facts is the same as a random guess model while being highly consistent. This is because the base Llama2-13B model classifies each fact (both sides of $\leftrightarrow$) as true, resulting in a consistent yet inaccurate fact-checker. Upon fine-tuning, the accuracy on both conjunctive and disjunctive fact-types improves significantly, while retaining a similar performance in logical consistency. Therefore, our experiments demonstrate the generality of applying supervised fine-tuning in improving both logical consistency and accuracy in logical fact checking across multiple fact types, including De Morgan’s law.
>
> In the final version of the paper, we shall include our aforementioned experimental results that fine-tuned LLMs also exhibit improved accuracy and consistency with respect to De Morgan’s law. We hope that this new set of experimental results and explanations will increase your support for our paper. We are looking forward to your feedback and whether we can provide any additional details to clarify your concerns.

---

### Official Review · Reviewer_pZwi · 2024-11-04

**Soundness:** 3
**Presentation:** 3
**Contribution:** 3
**Rating:** 5
**Confidence:** 3

**Summary:**

This paper presents a method for bench marking LLMs for logical consistency and ways to improve the logical consistency of LLMs through a supervised fine tuning using the fine tuning data. Paper shows how LLMs perform on different logical consistency checks with triples given a context to answer some logical queries and show the effectiveness of improving them with SFT. They propose three benchmark datasets based on three different knowledge bases and show how SFT improves across benchmarks.

**Strengths:**

Overall benchmarks for checking logical consistency for a given triple context and logical query are good and can help in evaluating logical consistency of LLMs.
Empirical results showing zero shot/prompt and fine tuning results show the improvements across benchmarks.

**Weaknesses:**

Experiments are more on the benchmark created and not sure how well they translate to real world situations or places where queries are in textual nature and does he supervised model still be able to generalize to textual data as opposed to triple context?

**Questions:**

After SFT, the models still perform as good on the original language benchmarks as the original model?
Was the goal of the peper to produce a model that does logical consistency in KG RAG kind of setting or in general to improve the logical consistency of LLMs ? if its later I don't see any experiments around other evaluation to show that.

---

> ### Author Response · Authors · 2024-11-20
> **Rebuttal**
>
> Thank you for appreciating our work and suggesting additional experiments. We consider all suggestions and present the summary of additional results in the rebuttal.
>
>
>
>
> > Experiments are more on the benchmark created and not sure how well they translate to real world situations or places where queries are in textual nature and does the supervised model still be able to generalize to textual data as opposed to triple context?
>
>
> Indeed, the benchmarks that we created are more structured than natural text queries. Following the reviewer's suggestion, we experiment with a widely known real-world fact-checking benchmark containing textual facts: FEVER (Fact Extraction and VERification) [1].
>
>
> - Dataset: FEVER consists of 185,445 claims generated by altering sentences extracted from Wikipedia, which are subsequently verified without knowledge of the sentence they were derived from. The claims are classified as supported, refuted, or not-enough-info. In experiments, we consider supported and refuted claims as simple query $p$ and manually create natural language negation of each claim ($\neg p$) to assess for logical consistency. Our augmented FEVER dataset contains 580 supported claims and 220 refuted claims, in a total of 580 + 220 = 800 simple claims. We split the 800 facts into 400-200-200 for training, validation, and test, respectively, to perform supervised fine-tuning.
>
>
> - Experiment:
> We adopt the Gemma-2B-it model as a proof of concept. In order to create the context, we retrieve 5 most relevant claims to the given query claim using vector embedding retrieval method. The following table shows that our method (of improving logical consistency) also works on real world situations or places where both queries and context to answer the queries are in natural text.
>
>
> |  | **Accuracy  (Gemma-2B)** | **Accuracy  (Fine-tuned Gemma-2B)** | **Consistency  (Gemma-2B)** |  **Consistency  (Fine-tuned Gemma-2B)** |
> |---|---|---|---|---|
> | **Simple fact** | 0.73 | **0.95** |  0.53 | **0.96** |
>
>
> In summary, the LLM (prior to fine-tuning) still lacks logical consistency and accuracy on natural text facts. Our supervised fine-tuning improves both logical consistency and accuracy of the LLM and hence generalizes to textual data.
>
>
> - Limitations: We are unable to find a dataset containing complex facts in natural text such that we can logically manipulate the facts to assess logical consistency. One viable option is to synthetically construct complex claims by combining claims from FEVER dataset; however manual curation of such a dataset takes time and resources we lack in the limited rebuttal window. For instance, we try to create such a dataset using stronger LLMs (such as Llama3 or GPT) but find that the generated claims still need to be checked manually, one by one, to ensure the fidelity of the claims.
> We have added these results in Appendix F of the revised manuscript.

---

> ### Author Response · Authors · 2024-11-20
> **Rebuttal Continued**
>
> > After SFT, the models still perform as good on the original language benchmarks as the original model?
>
>
> During rebuttal, we conducted an evaluation of SFT models on four general language benchmarks: SCIQ [2], Lambada [3], MMLU [4], and Hellaswag [5]. Among them, logical consistency-based fine-tuning improves the benchmark accuracy on SCIQ (from 0.926 to 0.94 in Gemma-2B and 0.94 to 0.945 in Llama2-7B), which is a science exam question-answer benchmark. In addition, on the Lambada benchmark, the performance increases in the initial few epochs. In the rest of the benchmarks, MMLU and Hellaswag, increasing fine-tuning epochs results in a monotonic decrease in benchmark accuracy with epochs.
>
>
> Therefore, consistently across multiple models like Gemma and Llama, consistency-based fine-tuning is helpful in the scientific domain benchmark, while a performance degradation may be observed in other multi-task abilities with excessive fine-tuning. As such, as is evident from our experiments, we suggest applying consistency-focused SFT for a few epochs before the model overfits and loses its generalization performance on other benchmarks.
>
>
> The detailed results are added in Appendix G of the revised manuscript.
>
>
>
>
> > Was the goal of the paper to produce a model that does logical consistency in KG RAG kind of setting or in general to improve the logical consistency of LLMs ? if its later I don't see any experiments around other evaluations to show that.
>
>
> We specifically target the KG RAG setting for two reasons: (1) Real-world KGs store a large number of highly-curated facts; thus, they provide a test bed for LLMs’ logical consistency assessments. We construct KG facts-based propositional fact-checking queries, provide KG contexts to LLMs, and assess the consistency of their responses under logical manipulation. (2) For our supervised fine-tuning to improve LLMs’ consistency, we have the availability of abundant training data with ground-truth labels from KGs.
>
>
> To extend to general consistency improvement, we anticipate our experiments can be a guide. For instance, a generally consistent LLM needs to be accurate (a sufficient condition) on constituent atomic queries corresponding to the base complex queries. Such a dataset can be constructed by modifying any standard instruction fine-tuning dataset. However, the constraint is that the fine-tuning dataset needs to be formulated as a propositional logic-based query so that we can manipulate the query logically to evaluate consistency. We mentioned this constraint in paragraph 2 of Section 6 (conclusion) in our original manuscript.
>
>
>
> ` References: `
>
>
> [1] Thorne, James, et al. "FEVER: a large-scale dataset for fact extraction and VERification." arXiv preprint arXiv:1803.05355 (2018).
>
>
> [2] Welbl, Johannes, Nelson F. Liu, and Matt Gardner. "Crowdsourcing multiple choice science questions." arXiv preprint arXiv:1707.06209 (2017).
>
>
>
>
> [3] Paperno, Denis, et al. "The LAMBADA dataset: Word prediction requiring a broad discourse context." arXiv preprint arXiv:1606.06031 (2016).
>
>
>
>
> [4] Hendrycks, Dan, et al. "Measuring massive multitask language understanding." arXiv preprint arXiv:2009.03300 (2020).
>
>
>
>
> [5] Zellers, Rowan, et al. "Hellaswag: Can a machine really finish your sentence?." arXiv preprint arXiv:1905.07830 (2019).

---

> ### Author Response · Authors · 2024-11-23
> **Looking forward to hearing from you!**
>
> Dear Reviewer pZwi,
>
> We wonder whether we can provide any additional information to clarify your questions/concerns about real world situations where queries are in textual nature and our SFT-based models' performance on the original language benchmarks.
>
> Thank you again for your constructive comments and we are looking forward to your feedback!
>
> Paper 3993 Authors

---

> ### Comment · Area_Chair_CGA1 · 2024-11-25
> **Reminder: Rebuttal Deadline for ICLR 2025**
>
> Dear Reviewer pZwi,
>
> As the rebuttal deadline approaches, please kindly check the papers' discussion threads and respond to the authors' rebuttals. If you haven't had a chance to respond yet, I’d greatly appreciate your input soon. Your insights are invaluable to the authors and the review process.
>
> Thank you for your effort and support!
>
> Best regards,
>
> Area chair

---

### Author Response · Authors · 2024-11-20
**General comment and summary of changes**

We thank the reviewers for their valuable feedback to improve our work. We have uploaded a revised draft in which we made changes (in blue color) to address the concerns of the reviewers. In the following, we summarize the key changes.

- **Limitations and Broader Impacts** (Appendix A):  We have extended the discussion on Limitations of the paper, in particular, regarding (1) the general reasoning ability of LLMs and (2) our paper’s focus on binary responses (Reviewers Eq6F, 7P9X)

- **Dynamically Varying Contexts** (Appendix E, lines 1165-1171): We have discussed that the “BFS + Relation hierarchy” retrieval method is computationally efficient than embedding-based retrieval methods in the case of dynamic KGs. (Reviewer 7P9X)


- **Empirical Evaluation on Natural Text Dataset** (Appendix F): We have empirically evaluated our method on FEVER dataset, a widely known real-world fact-checking benchmark containing textual facts. Experiments show that our supervised fine-tuning improves both the logical consistency and accuracy of the LLM on natural text dataset, thereby suggesting that the proposed method also generalizes to textual data.  (Reviewer pZwi)


- **Performance of the Fine-tuned Model on Language Benchmarks** (Appendix G): We have conducted an evaluation of the fine-tuned SFT model on four general language benchmarks: SCIQ, Lambada, MMLU, and Hellaswag. We found that the accuracy improved on the scientific question benchmark SCIQ and open-ended cloze task dataset Lambada. We have discussed remedies that may help improve the performance on other benchmarks. (Reviewer pZwi)


- **Experimental Results with the MiniCheck Baseline** (Appendix H): We have empirically compared the baseline MiniCheck-FT5 and our fine-tuned Llama2-7B model on the Freebase dataset. We found that our fine-tuned Llama2-7B shows better accuracy and consistency than MiniCheck-FT5. This highlights the importance of our supervised fine-tuning on propositional logic queries. (Reviewer JETn)

- **Experiments with Larger Models** (Appendix I): In an effort to evaluate the need for supervised fine-tuning on larger-sized models, we have evaluated Llama2-70B on simple facts, complex facts, and logic rules. We found that despite having more parameters than the corresponding 7B and 13B versions of LLaMA2, Llama2-70B does not always improve logical consistency in all three query types, demanding further supervised fine-tuning or instruction prompting. We further found that 70B model demonstrates better instruction prompting capability than the lower size models, while improving accuracy and logical consistency. (Reviewer JETn)

- **Grammatical Error** (line 346): We have corrected the grammatical error at line 346 in section 4.1 (Reviewer Eq6F)

Below, we address the questions and concerns specific to each reviewer in further detail.

---

### Meta-Review · Area_Chair_CGA1 · 2024-12-22

**Metareview:**

Summary of the paper: This paper addresses the critical issue of logical consistency in LLMs, particularly in the context of answering complex queries derived from propositional logic and knowledge graphs. This paper highlights a gap in current assessments, which often focus on simple paraphrasing rather than deeper logical reasoning. They present a method to benchmark LLMs for logical consistency through three proposed datasets (FreebaseLFC, NELLLFC, and WikiLFC) that evaluate how well models maintain consistency when answering logically equivalent queries. To enhance logical consistency, this paper advocates for SFT over prompting, demonstrating its effectiveness in improving model performance across the benchmarks. Overall, the paper emphasizes the need for rigorous evaluation and improvement of LLMs in logical reasoning tasks, particularly in applications such as fact-checking.

Strengths of the paper:
- Clear Presentation: In general, this paper is well-written, presenting complex concepts in an accessible manner.
- Useful Resources: This paper establishes a definition of logical consistency and introduces three new datasets (FreebaseLFC, NELLLFC, WikiLFC) for fine-tuning and evaluation.
- Comprehensive Experiments: The study employs multiple experimental setups, comparing zero-shot instruction prompting with supervised fine-tuning, revealing limitations in existing LLMs' logical consistency. Results show significant improvements in the logical consistency of LLMs when applying supervised fine-tuning techniques.

Weaknesses of the paper:
- Benchmark Limitations (Reviewer pZwi): Experiments are heavily reliant on created benchmarks, raising concerns about their applicability to real-world scenarios, particularly with textual queries and the generalizability of supervised models.
- Dynamic Context Effectiveness (Reviewer 7P9X): The methods for extracting relevant KG context via BFS and embedding-based retrieval lack clarity on their effectiveness in dynamically varying real-world situations, indicating a need for further exploration.
- Limitation of Reasoning Type (Reviewers Eq6F, 7P9X): The focus on binary responses could restrict the method's applicability in complex domains, suggesting a need to explore multi-class logic and non-binary answer consistency for broader applications.
- More Analysis are Needed (Reviewer JETn, pZwi): Including using larger LLMs, more baselines and more benchmarks to provide more empricial evidence to justify the effectiveness of the proposed method.

Reason for the decision: After considering the rebuttal, I believe that the authors have adequately addressed most of the concerns raised, as detailed in the additional comments in the reviewer discussion box. Reviewers JETn and 7P9X support accepting this paper, while reviewers pZwi and Eq6F recommend rejection. Reviewers pZwi and Eq6F did not respond to the authors' last comments. Upon reviewing the authors' responses, I find that they have addressed, at least in part, the concerns raised by these reviewers. After careful consideration, I am leaning toward accepting this paper due to its proposed logical consistency, which I believe is an important perspective that is currently somewhat overlooked by the research community. Additionally, the paper contributes valuable resources, including three datasets and an effective SFT method. It has the potential to engage the broader community and make an impact.

**Additional Comments On Reviewer Discussion:**

Thanks to the authors summarizing the rebuttal, I think most of the issues raised by the reviewers have been addressed as follows:
- Benchmark Limitations (Reviewer pZwi): The authors have empirically evaluated their method on FEVER.
- Dynamic Context Effectiveness (Reviewer 7P9X): The authors discussed the “BFS + Relation hierarchy” retrieval method in the Appendix.
- Limitation of Reasoning Type (Reviewers Eq6F, 7P9X): The authors have extended the discussion on the Limitations of the paper, in particular, regarding (1) the general reasoning ability of LLMs and (2) our paper’s focus on binary responses
- More Analysis are Needed (Reviewer JETn, pZwi): Fine-tuned model on four general benchmarks, compared with MiniCheck-FT5 and evaluated with Llama2-70B.

---

### Decision · Program_Chairs · 2025-01-22

Accept (Poster)